# Private Federated Learning using Preference-Optimized Synthetic Data

**Charlie Hou** [* 1]  **Mei-Yu Wang** [* 2]  **Yige Zhu** [*]  **Daniel Lazar** [3]  **Giulia Fanti** [1]

## Abstract

In practical settings, differentially private federated learning (DP-FL) is the dominant method for training models from private, on-device client data. Recent work has suggested that DP-FL may be enhanced or outperformed by methods that use DP synthetic data (Wu et al., 2024; Hou et al., 2024). The primary algorithms for generating DP synthetic data for FL applications require careful prompt engineering based on public information and/or iterative private client feedback. Our key insight is that the private client feedback collected by prior DP synthetic data methods (Hou et al., 2024; Xie et al., 2024) can be viewed as an RL reward. Our algorithm, Policy Optimization for Private Data (POPri) harnesses client feedback using policy optimization algorithms such as Direct Preference Optimization (DPO) to fine-tune LLMs to generate high-quality DP synthetic data. To evaluate POPri, we release LargeFedBench, a new federated text benchmark for uncontaminated LLM evaluations on federated client data. POPri closes the gap in performance between the fully-private and non-private settings by up to 58%, compared to 28% for prior synthetic data methods, and 3% for state-of-the-art DP federated learning methods. The code and data are available at https://github.com/meiyuw/POPri.

## 1. Introduction

Many important machine learning (ML) applications feature sensitive datasets that are distributed across client devices (e.g. mobile devices). Such ML models are often hosted on client devices. These *on-device* models offer privacy, latency, and storage benefits relative to centrally-hosted mod-els. Examples include Google's GBoard (Hard et al., 2019; Xu et al., 2023b; Wu et al., 2024) and Apple's mobile automatic speech recognition system (Paulik et al., 2021). Today, federated learning (FL) is the most widely-used approach in practice for learning on-device models; it trains models locally on user devices and aggregates model updates on a central server (McMahan et al., 2017b). FL protects the privacy of client data in part by adopting differentially private (DP) (Dwork, 2006) optimization techniques, a combination we refer to as DP-FL (McMahan et al., 2017b; Kairouz et al., 2021b; Nguyen et al., 2022; Xu et al., 2023a).

With breakthroughs in large language model (LLM) capabilities (Anil et al., 2023; Team et al., 2023; Achiam et al., 2023; Guo et al., 2025) several research teams have used LLMs to better train models on private client data. A common strategy applies standard optimization algorithms (e.g., DP stochastic gradient descent, DP-SGD (Abadi et al., 2016)) to fine-tune models on private client data (Kurakin et al., 2023; Charles et al., 2024). These approaches have an important limitation in the on-device setting: most LLMs today are too large to fit on client devices, let alone train on them (Radford et al., 2019; Touvron et al., 2023).

To sidestep the size issue, Wu et al. (2024); Hou et al. (2024) view the problem of learning from distributed, private client data (partially) as a DP synthetic data problem. These approaches use LLM-assisted workflows to generate privacy-preserving synthetic data, similar to client data, at the server; then they train the on-device model *at the server* on the synthetic data. This avoids storing the LLM on client devices.

In more detail, Wu et al. (2024) use prior public information about the clients to create LLM-generated synthetic data for pretraining. For example, for their Google GBoard virtual keyboard application, they use prompts like "Imagine you are a [GENDER] of age [AGE]. Write some examples of chat messages." to generate synthetic samples. This prompt was designed entirely using prior qualitative information about the data on client devices. However, prior information may not always be available. Moreover, this prompt was not refined based on clients' realized data, which could limit the relevance of the resulting synthetic data.

PrE-Text (Hou et al., 2024) instead uses Private Evolution (PE) (Lin et al., 2023; Xie et al., 2024; Lin et al., 2025b) to learn prompts that are relevant to client data. PE itera-

---

[*]Equal contribution [1]Department of ECE, Carnegie Mellon University, Pittsburgh, PA [2]Pittsburgh Supercomputing Center, Pittsburgh, USA [3]Coldrays, Tucson, AZ. Correspondence to: Charlie Hou <hou.charlie2@gmail.com>.

*Proceedings of the 42nd International Conference on Machine Learning*, Vancouver, Canada. PMLR 267, 2025. Copyright 2025 by the author(s).

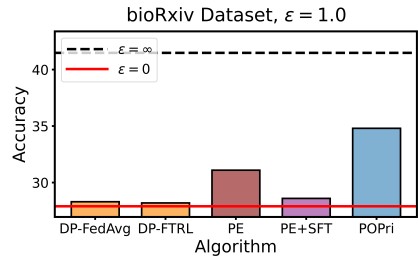

*Figure 1.* **Left:** Private Evolution (PE)-based techniques. Clients generate low-dimensional statistics which summarize the similarity of the synthetic data to their private samples. These are privately aggregated to refine the synthetic data generation for future iterations. Traditional PE (brown) uses a prompt-based method. POPri (blue) improves a naive fine-tuning method (PE+SFT, purple) by fine-tuning the LLM using *policy optimization* rather than fine-tuning directly on aggregated client feedback. **Right:** Next-token prediction accuracy on the bioRxiv dataset at privacy level $\epsilon = 1$. POPri closes the accuracy gap between the fully-private and non-private settings by 58%, compared to 23% for prior synthetic data methods, and 3% for DP federated learning methods.

tively sends synthetic data samples to clients for feedback; each client privately measures the closeness of synthetic samples to their own data, discarding irrelevant samples. It returns this feedback to the central server, which crafts a new prompt based on the most relevant synthetic samples. Finally, an LLM uses the generated synthetic data to fine-tune a downstream model. This method of utilizing LLMs for on-device learning has some shortcomings: (1) it relies entirely on *prompting* to teach the LLM to generate relevant synthetic data, which may not be as effective as fine-tuning the weights. (2) It discards irrelevant samples, which may themselves contain valuable information, as shown in reinforcement learning with human feedback (RLHF) (Ouyang et al., 2022).

In this paper, we demonstrate how to better utilize LLMs for on-device learning: we propose **POPri** (Policy Optimization for Private Data), an algorithm that reformulates synthetic data-based approaches for private learning as an LLM policy optimization problem. In POPri, we directly fine-tune an LLM's weights to improve the (DP-noised) similarity scores between generated synthetic data and private client samples. The fine-tuned LLM is used to generate synthetic data, which is used to train a downstream model.

**Contributions.** In summary, our contributions are:

(1) We propose POPri, a novel method that casts private learning under the synthetic data framework as an LLM policy optimization problem. Prior work in this space relied on PE, which uses client feedback exclusively to generate new prompts (Hou et al., 2024; Xie et al., 2024). We alter this feedback to instead provide client *rewards*, and subsequently exploit recent advances in policy optimization (Rafailov et al., 2023). This recasting allows us to more effectively exploit the capabilities of LLMs for on-device learning problems.

(2) We create and maintain LargeFedBench, a new uncontaminated benchmark of federated client data separated by client for the era of LLMs. The datasets in this benchmark consist of: (1) congressional records in English-speaking countries, and (2) abstracts from bioRxiv, collected starting in April 2023. To our knowledge, this is the first dataset that provides researchers with both (a) over 1,000 clients (congressional records contains 134k clients and bioRxiv contains 57k as of August 2024), and (b) regular updates, allowing researchers to easily filter data to avoid contaminated evaluations (Magar & Schwartz, 2022; Zhou et al., 2023; Yang et al., 2023; Roberts et al., 2023).

(3) We demonstrate the utility of POPri on this new benchmark set of datasets, as well as two central (i.e., the setting where all data is present on the server and no server-client communication is needed) DP benchmarks from prior work (Yu et al., 2023; Xie et al., 2024). Across all datasets and tasks (we consider next token prediction and text classification), POPri achieves the best downstream metrics. For example, Figure 1 shows that on our bioRxiv dataset at a privacy level of $\epsilon = 1.0$, POPri outperforms PE-based algorithms by 6 full percentage points, and closes the gap between fully private and non-private baselines by over 58%, compared to 23% for PE. It outperforms DP-FL-based methods by even more. Additional experimental details, results, and ablations are provided in Section 5.

## 2. Problem Statement and Background

### 2.1. Problem Statement

We consider a set $\mathcal{S}$ of clients, $\mathcal{S} = \{S_1, \ldots, S_n\}$, where $S_i = \{s_1^{(i)}, \ldots, s_{m_i}^{(i)}\}$ denotes the private text data of client $i \in [n]$, and $m_i$ denotes the number of text samples held by client $i$. We consider the partial participation setting, where only a subset of clients can participate in communication with the server at any point in time (Kairouz et al., 2021a;

McMahan et al., 2017b), which is consistent with practical private on-device learning deployments. We assume $L$ clients participate in each round $t \leq T$ and denote this set $\mathcal{S}^t$. We do not assume an *a priori* upper bound on $m_i$. A central server is given a pre-trained downstream model $\Phi$, which it wants to align with the private client data $\mathcal{S}$. We call the aligned downstream model $\tilde{\Phi}$. In the process of learning $\tilde{\Phi}$, the server may make use of a pre-trained public LLM $\Psi$. We observe that $\Psi$ and $\Phi$ are different models in general; we will assume the server has access to the weights of both $\Phi$ and $\Psi$. The server is subject to two restrictions: (1) client data cannot leave client devices, and (2) the final model $\tilde{\Phi}$ must protect user-level differential privacy (DP):

**User-level (distributed) differential privacy (DP).** We say two datasets $\mathcal{S}$ and $\mathcal{S}'$ are *neighboring* if they differ in at most one client's data. That is, there exists an $i \in [n]$ such that for all $j \neq i$, $S_j = S_j'$. A randomized mechanism $\mathcal{M}$ is $(\epsilon, \delta)$-DP if, for any pair of neighboring datasets $\mathcal{S}, \mathcal{S}'$ that differ by an entire client's data and any possible output set $E$, it holds that $\Pr[\mathcal{M}(\mathcal{S}) \in E] \leq e^\epsilon \Pr[\mathcal{M}(\mathcal{S}') \in E] + \delta$. The post-processing property of a DP mechanism ensures that any data-independent transformation applied to its output preserves the same DP guarantees. (Dwork, 2006; Dwork & Roth, 2014).

We also evaluate on central DP baselines, so we define central DP below; in that case the final model $\tilde{\Phi}$ should protect central DP:

**Central (example-level) differential privacy (DP).** We say two datasets (both fully present on the server) $S = \{s_1, \ldots, s_m\}$ and $S' = \{s_1', \ldots, s_m'\}$ are *neighboring* if they differ in at most one example's data. That is, there exists an $i \in [m]$ such that for all $j \neq i$, $s_j = s_j'$. A randomized mechanism $\mathcal{M}$ is $(\epsilon, \delta)$-DP if, for any pair of neighboring datasets $\mathcal{S}, \mathcal{S}'$ that differ by one sample and any possible output set $E$, it holds that $\Pr[\mathcal{M}(S) \in E] \leq e^\epsilon \Pr[\mathcal{M}(S') \in E] + \delta$.

**Goal.** The server seeks an algorithm to optimize the downstream performance (in our paper, this is either next token prediction accuracy or text classification accuracy) of $\tilde{\Phi}$ on a test set of private data, subject to an $(\epsilon, \delta)$-DP constraint.

## 2.2. Related Work

There are two main approaches for learning on private data.

**DP optimization-based approaches.** In natural language processing (NLP) tasks with privacy constraints, DP optimization algorithms (e.g., DP-SGD (Abadi et al., 2016)) are often used to fine-tune massively pretrained LLMs on private data (Bommasani & Schofield, 2019; Kurakin et al., 2023; Charles et al., 2024). However, in settings where client data cannot leave client devices due to privacy con-

cerns, central servers cannot conduct this private fine-tuning.

An alternative approach is to train models directly *on client devices*, using a server to coordinate information exchange between clients; in DP federated learning (DP-FL) (McMahan et al., 2017b; Kairouz et al., 2021a), (small) model weights are iteratively sent to clients for on-device DP optimization. DP-FL has struggled to keep up with the growing size of LLMs; many LLMs cannot be stored or trained on client devices (Collins et al., 2023). Recent work explores how to train LLMs in the DP-FL framework. Proposed approaches include training only subsets of parameters (Charles et al., 2023), as well as memory-efficient zero-order optimization (Zhang et al., 2024; Malladi et al., 2023). However, these methods still require the storage of the entire model on-device, limiting their practicality.

**Synthetic data-based approaches.** An alternative approach to DP optimization involves generating private synthetic data using LLMs, followed by directly fine-tuning downstream models. Synthetic data can be generated on the server side, which bypasses client-side hardware constraints. The post-processing property of DP also implies that DP synthetic data can be used repeatedly without incurring additional privacy loss (Yue et al., 2023a). In the centralized DP setting (where the server is trusted to gather all the data, as opposed to our private on-device setting), prior studies have shown that training downstream models on DP synthetic text achieves performance comparable to privately training on real data (Yue et al., 2023a; Mattern et al., 2022; Xie et al., 2024). In the **private on-device** setting, Hou et al. (2024) show that fine-tuning a small model on user-level DP synthetic text data on the server side can actually *outperform* DP-FL, with a significant reduction in communication and computation cost. Similarly, Wu et al. (2024) show that pretraining an FL model on private synthetic data can improve the final outcome of DP-FL.

One approach for generating synthetic text data is to fine-tune an LLM (with DP-SGD) on private data (Kurakin et al., 2023; Yu et al., 2024) and then using the LLM to generate synthetic data. However, client hardware constraints render this approach infeasible on-device. Recent works have relied instead on privacy-aware prompt engineering to generate synthetic data (Wu et al., 2024; Xie et al., 2018; Hou et al., 2024). An important framework by Lin et al. (2023; 2025a) called **Private Evolution** (PE) is the basis for several competitive DP synthetic text algorithms, including Aug-PE (Xie et al., 2024) and PrE-Text (Hou et al., 2024). Roughly, these algorithms use the public LLM $\Psi$ to generate synthetic data, score each synthetic data according to its closeness to the client data, and discard synthetic data with low scores. The surviving synthetic data are used as in-context examples for $\Psi$ to generate synthetic data. In concurrent work to ours, Zou *et al.* extend the PE framework to

generate synthetic data from multiple pretrained language models (LMs), and present "good" and "bad" responses to the LMs in the next round for in-context learning (Zou et al., 2025). Private Evolution may sacrifice data quality in two ways: First, it uses in-context learning, which is often less effective than fine-tuning (Mosbach et al., 2023). Second, discarding low-score synthetic data may lose useful information (Ouyang et al., 2022). We address both by turning the DP synthetic generation problem into an LLM policy optimization problem.

## 3. POPri

The core idea of POPri (**P**olicy **O**ptimization for **Pri**vate Data) is a natural reformulation of private on-device learning from synthetic data as an LLM policy optimization problem, which enables the use of powerful LLM alignment methods like DPO (Rafailov et al., 2023). In this section, we detail the POPri design principles and algorithm. POPri's design is based on two related questions.

### 1. What client feedback should we collect for fine-tuning? Three natural options arise:

(1) *DP Data.* Clients could directly transmit DP synthetic data samples for fine-tuning, e.g., using a method like DP-Prompt (Utpala et al., 2023). DP-Prompt uses an LLM to summarize text at a temperature specified by the desired DP $\epsilon$ level. However, DP text cannot be aggregated into a single statistic, which prevents the use of secure aggregation (Bonawitz et al., 2016); this increases the noise needed to reach a given DP guarantee. As such, prior work has shown that DP-Prompt is not competitive with other private on-device learning methods (Hou et al., 2024). We favor aggregation-compatible representations of client data, such as summary statistics or model parameters.

(2) *DP Model Parameters.* A second alternative is to send the parameters of either the LLM $\Psi$ or the downstream model $\Phi$ to the client and train on the private samples with DP-SGD (Abadi et al., 2016). These parameters are compatible with secure aggregation (Bonawitz et al., 2016), which makes more efficient use of DP budget. However, $\Psi$ cannot be sent to clients because of client storage constraints. Sending $\Phi$ is the DP-FL approach, which is one of our baselines.

(3) *DP Statistics.* Finally, we could collect low-dimensional statistics capturing the quality of synthetic data samples. In PE, the server generates $K$ synthetic data samples (Xie et al., 2024; Hou et al., 2024), and each client computes a histogram counting how often each of the private samples is closest to one of the $K$ samples. This $K$-dimensional histogram can be made DP by adding (comparatively) little noise, and it is amenable to secure aggregation (Xie et al., 2024; Hou et al., 2024). We view such low-dimensional

statistics as the most promising option, as they have lower communication and storage costs, and they make better use of the privacy budget. *In a departure from PE, we design the low-dimensional statistics collected by POPri to enable building a preference dataset.* We ask the server to generate $J$ samples from each of $K$ prompts; each client then scores the $K \times J$ samples according to how well they represent the client's data, and the server aggregates the scores for all the synthetic samples. Using these scores, the server can construct a "higher scoring response" and "lower scoring response" pair (a "preference pair") for each of the $K$ prompts. The benefit of this new design ties directly to the next question.

### 2. How should we use client feedback?

Given a vector summarizing the quality of synthetic data samples, how should we use it? A few options arise:

(1) *In-Context Learning.* We could use the highest-scoring synthetic samples as in-context examples to prompt the LLM $\Psi$. This is the PE approach (Hou et al., 2024; Xie et al., 2024). However, in-context learning typically performs worse than finetuning-based approaches (Mosbach et al., 2023), and we find experimentally that POPri outperforms Private Evolution (PE) (Figure 1, Table 1).

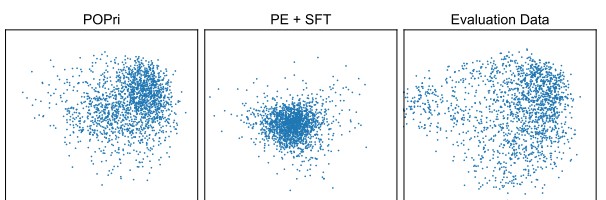

| POPri | PE + SFT | Evaluation Data |

*Figure 2.* 2-PCA visualization of synthetic data from POPri and PE+SFT, and evaluation data. We see that POPri's synthetic data distribution (**left**) is much closer to the evaluation data distribution (**right**) than the PE+SFT synthetic data distribution (**middle**). Naive fine-tuning with SFT on PE-generated synthetic data does not make best use of client feedback.

(2) *Supervised Fine-Tuning (SFT).* One could directly fine-tune the LLM $\Psi$ on the highest scoring samples using next-word-prediction loss. This is analogous to the SFT baseline evaluated in the RLHF (Ouyang et al., 2022) and DPO (Rafailov et al., 2023) papers, which showed that RLHF and DPO outperform SFT. The reason is that the highest scoring samples–while better than the low-scoring samples–are not perfect responses to the prompt. The SFT loss trains the LLM to treat high-scoring samples as perfect responses, which is misaligned with the LLM's task. Empirically, we see that this approach (PE+SFT) produces synthetic data that is not representative of the private data (Figure 2) and has poor downstream performance (Table 1).

(3) *Policy Optimization (PO).* Policy optimization-based

**Algorithm 1** POPri

1: **Input:** Clients private data $\{S_i\}_{i \in [n]}$, Number of rounds $T$, Number of generated samples $N_{\text{syn}}$, Noise multiplier $\sigma$, LLM $\Psi$, embedding model $\Gamma$, base prompt $\eta$, participating clients in each round $\mathcal{S}^t$, "rejected" index $\ell$, random prompt generator $\Lambda(\cdot)$, number of clients sampled $L$
2: **Output:** Synthetic data $S_{syn,T+1}$
3:
4: All clients $i \in [n]$ embed private samples, $E_i = \Gamma(S_i)$
5: Server initializes LLM $\Psi_1 = \Psi$
6: **for** $t \leftarrow 1 \dots T$ **do**
7:   **Server:**
8:   Initialize the response vector $R = \emptyset$
9:   **for** $k \leftarrow 1 \dots K$ **do**
10:     Generate prompt $\eta_k = \Lambda(\eta)$,
11:     Generate $J$ responses $R_{kj} = \Psi_t(\eta_k), j \in [J]$
12:   **end for**
13:   Send embeddings $E_{syn,t} = \{\Gamma(R_{kj})\}_{k \in [K], j \in [J]}$ to all clients in $\mathcal{S}^t$
14:
15:   **Client $i \in \mathcal{S}^t$:**
16:   $\text{Scores}_{i,t} \leftarrow \texttt{SIMILARITY}(E_{syn,t}, E_i)$
17:   Send $\text{Scores}_{i,t} + \mathcal{N}(0, \sigma^2 I/L)$ to Server
18:
19:   **Server:**
20:   Secure aggregate scores: $\text{Scores}_t = \frac{1}{L}\sum_{i \in \mathcal{S}^t} \text{Scores}_{i,t}$
21:   Set $P[k,j]$ as the $j$-th highest score response for prompt $\eta_k$, according to $\text{Scores}_t$
22:   Initialize preference dataset $\mathcal{P}_t = \emptyset$
23:   **for** $k \leftarrow 1 \dots K$ **do**
24:     Select positive synthetic sample: $\mathcal{P}_t[k,1] = P_t[k,1]$
25:     Select negative synthetic sample: $\mathcal{P}_t[k,2] = P_t[k,\ell]$
26:   **end for**
27:   Fine-tune: $\Psi_{t+1} \leftarrow \text{DPO}(\Psi_t, \{\eta_k\}_{k \in [K]}, \mathcal{P}_t)$
28: **end for**
29: **Server:**
30: Output final synthetic data $S_{syn,T+1}$ from $\Psi_T$

---

methods like DPO (Rafailov et al., 2023) instead directly optimize the LLM to produce higher-scoring samples (where the score can be defined by the user of the algorithm). In other words, they are designed to directly make use of the low-dimensional scores we collect from client feedback. Hence, we expect such methods to produce higher quality synthetic data, as evaluated on downstream tasks.

### 3.1. POPri Algorithm

Pseudocode can be found in Algorithm 1. We highlight the algorithmically new steps (that differ from PE) in blue .

**1. Synthetic sample generation.** We generate $K$ prompts (details in Appendix B.2). A prompt is generated by randomly sampling three samples from $\Omega$ and prompting LLaMA-3-8B (Touvron et al., 2023) to generate a fourth sample given the first three samples as examples. The exact

prompt is given in Appendix B. For each of the $K$ prompts, we generate $J$ synthetic samples (by running the prompt independently $J$ times). In total, the server generates $K \times J$ synthetic samples, embeds them using a small sentence embedding model $\Gamma$ and sends the embeddings to every client in $\mathcal{S}^t$, i.e., the clients sampled in round $t$.

**2. Scoring synthetic data using DP client feedback.** Next, each client in $\mathcal{S}^t$ scores the synthetic samples. Specifically, each client calculates, for each of the $K \times J$ synthetic samples, its cosine similarity with each of the client's private samples, averaged over the client's samples (Algorithm 4). The use of cosine similarity differs from PE, which uses a nearest neighbors histogram (Lin et al., 2023; Hou et al., 2024; Xie et al., 2024)–using cosine similarity is critical to the performance of POPri as we found in our ablations (see Section 5.2). These similarities for every synthetic sample are arranged into a vector. We clip this vector to a norm of 1, which caps the contribution of each client (similar to how gradient updates are clipped per client in DP-FL (McMahan et al., 2017b)). Clipping is done primarily for privacy reasons, as we will elaborate later. Clipping also ensures that the contribution of clients with large amounts of data does not overwhelm the contribution of clients with small amounts of data. We then add $\mathcal{N}(0, \sigma^2 I/L)$ (where $I$ is the identity matrix of size $KJ \times KJ$) noise to the resulting vector to ensure DP ($\sigma^2$ controls the $(\epsilon, \delta)$). Finally, we aggregate scores via secure aggregation (Bonawitz et al., 2016), yielding a DP score for each synthetic sample that reflects its relevance to client data.

**3. LLM Policy Optimization.** The key insight of our paper is that by generating $J$ synthetic samples from $K$ prompts and scoring all of them using DP client feedback, we can create a preference dataset where for each of the $K$ prompts, we can assemble a "good sample" and a "bad sample". This design choice allows the usage of powerful LLM policy optimization algorithms (we choose DPO (Rafailov et al., 2023)) to finetune the LLM $\Psi$. In detail, each of the $K$ prompts have $J$ synthetic samples which are ranked according to the scores we gathered. Then for each of the $K$ prompts, we set the highest scoring sample as the "chosen sample" and the $\ell$-th highest scoring sample as the "rejected sample". This resulting preference dataset can then be passed, along with the LLM $\Psi$, into the DPO preference optimization loss (Rafailov et al., 2023):

$$\min_{\Psi} \mathbb{E}_{\substack{x,y_\omega \\ y_r}} \left[ -\log s \left( \tau \log(\frac{\Psi(y_\omega|x)}{\Psi(y_r|x)}) - \tau \log(\frac{\Psi_{\text{ref}}(y_\omega|x)}{\Psi_{\text{ref}}(y_r|x)}) \right) \right]$$

where $\Psi_{\text{ref}}$ a fixed checkpoint for the LLM (we use the public checkpoint of the LLM), $\tau$ is a parameter controlling deviation of $\Psi$ from $\Psi_{\text{ref}}$, $x$ is the prompt, $y_\omega$ is the chosen sample, $y_r$ is the rejected sample, $\Psi(y|x)$ is the probability

*Table 1.* Accuracy (%, ↑) of different algorithms on a variety of tasks and datasets (bioRxiv, Congress, PubMed are next-token-prediction accuracy, OpenReview is text classification accuracy). The highest accuracy across all methods is in **bold**. All standard deviation error bars are less than 0.5.

| Dataset | Method | Data Type | On-device Model | $\epsilon = \infty$ | $\epsilon = 7$ | $\epsilon = 1$ | $\epsilon = 0$ |
|---|---|---|---|---|---|---|---|
| bioRxiv | DP-FedAvg | Original | | | 29.0 | 28.3 | |
| | DP-FTRL | Original | | | 29.0 | 28.2 | |
| | PE | Synthetic | DistilGPT2 | 41.5 | 31.0 | 31.1 | 27.9 |
| | PE + SFT | Synthetic | | | 28.6 | 28.6 | |
| | POPri (ours) | Synthetic | | | **34.4** | **34.8** | |
| Congress | DP-FedAvg | Original | | | 29.1 | 29.0 | |
| | DP-FTRL | Original | | | 29.1 | 29.0 | |
| | PE | Synthetic | DistilGPT2 | 35.7 | 27.3 | 27.0 | 26.9 |
| | PE + SFT | Synthetic | | | 27.1 | 27.1 | |
| | POPri (ours) | Synthetic | | | **30.6** | **30.4** | |
| PubMed (Yue et al., 2023b) | PE | Llama-2-7b-chat-hf, Synthetic (2000) | BERT$_{\text{small}}$ | 47.6 | — | 27.5 | |
| | PE | Opt-6.7b, Synthetic (2000) | | | — | 27.9 | |
| | POPri (ours) | Synthetic (2000) | | | — | **29.4** | |
| OpenReview (Xie et al., 2024) | PE | Llama-2-7b-chat-hf, Synthetic (2000) | ROBERTA$_{\text{base}}$ | 50.8 | — | 37.0 | 32.0 |
| | PE | Opt-6.7b, Synthetic (2000) | | | — | 32.1 | |
| | POPri (ours) | Synthetic (2000) | | | — | **40.2** | |

of generating $y$ given $x$ for $\Psi$, and $s$ is the sigmoid function. The expectation is taken with respect to the empirical distribution (i.e. real samples). The DPO loss will be used to finetune $\Psi$ to generate more samples similar to the chosen sample and fewer like the rejected sample. To reduce GPU memory use, we use LoRA (Hu et al., 2021) on all the attention matrices and up/down projection matrices with a rank of 4, $\alpha = 8$. After fine-tuning over the $K$ prompts and preference pairs, we return back to step (2) and generate new synthetic data using the newly fine-tuned $\Psi$.

**4. Synthetic data generation for downstream tasks.** Using the final version of $\Psi$, we generate a large set of synthetic data $S_{syn,T+1}$ which is used to fine-tune $\Phi$ into $\tilde{\Phi}$. $\tilde{\Phi}$ is then sent to all the client devices, where they can perform inference without communicating information to the server.

**Privacy guarantees.** Because each client's vector is clipped to 1, and the only information revealed to the server (or any other party) is the aggregated vector, the sensitivity of the algorithm is 1. We add $\mathcal{N}(0, \sigma^2 I/L)$ noise to each client's vector, so the vector given to the server has noise $\mathcal{N}(0, \sigma^2 I)$, satisfying the Gaussian Mechanism with sensitivity 1. To calculate privacy, we can use a privacy accountant like `OPACUS.ACCOUNTANTS.ANALYSIS.RDP` (Yousefpour et al., 2021), and input $T$ (the number of rounds we run the algorithm, $q$ (the fraction of clients sampled per round), $\delta$, and set $\sigma$ to get the desired $\epsilon$ value.

## 4. LargeFedBench: A Federated Benchmark for LLM Evaluation

Today, the most widely-used evaluation datasets for federated learning of text models come from the work of Reddi et al. (2020); they include text from StackOverflow posts

and Shakespeare plays. These datasets pose two evaluation challenges: (1) They pre-tokenize inputs in a non-invertible way, which prevents researchers from using custom tokenizers adopted by several LLMs. (2) The datasets may lead to contaminated evaluations. As state-of-the-art LLMs have been trained on large swaths of the public internet, old public benchmark datasets may be in the training data of many LLMs (Magar & Schwartz, 2022; Zhou et al., 2023; Yang et al., 2023; Roberts et al., 2023). To our knowledge, one work proposes a benchmark dataset for federated LLMs (Ye et al., 2024). The datasets in this paper have at most 747 clients, which may be insufficient for simulating production use cases. Further, they do not explicitly avoid contamination.

We release **LargeFedBench**, a benchmark comprising two new datasets, Congressional Speeches and bioRxiv, for experiments over federated client data. These datasets (a) allow researchers to easily avoid contamination, and (b) provide enough distinct clients to simulate production settings.

**Congressional Speeches ("Congress")**[1] is a dataset of 134k speeches or debates scraped from congressional or parliamentary transcripts in the US, UK, and Canada. We treat each speech as a separate client, and samples are created as successive 64-token spans within the speech. **bioRxiv**[2] is a dataset of 57k abstracts, each of which we consider a client dataset of strings, scraped from biology papers. Samples are 64-token spans of the abstract. More details on the datasets are included in Appendix F.

A key feature of our datasets is that they are updated every

---

[1] https://huggingface.co/datasets/hazylavender/CongressionalDataset
[2] https://huggingface.co/datasets/hazylavender/biorxiv-abstract

*Figure 3.* Next-token prediction accuracy performance of four methods as a function of the number of clients sampled per round out of 10000. We see that across different client participation scenarios, POPri consistently performs the best.

6 months and sorted by date. Hence, researchers can easily select datasets that were generated after their model's knowledge cutoff date. In this paper, we use data from LargeFedBench published between the dates of April 2023 to August 2024 to avoid contamination with the latest LLM we evaluate our algorithms with, LLaMA-3-8B (AI@Meta, 2024)–which has a knowledge cutoff of March 2023.

## 5. Experiments

**Datasets and tasks.** For next token prediction accuracy, we evaluate POPri on the LargeFedBench datasets (**Congress** and **bioRxiv**), as well as **PubMed** (Yu et al., 2023; Xie et al., 2024) used in the evaluation of Private Evolution (Aug-PE) (Xie et al., 2024). PubMed contains abstracts of medical papers published between August 1-7, 2023 (details in Appendix D.2.2). For text classification, we evaluate POPri on **OpenReview** consisting of ICLR 2023 reviews published on November 5, 2022 which was used in the evaluation of Private Evolution (Aug-PE) (Xie et al., 2024). Note that PubMed and OpenReview are evaluations in the *central DP setting*, where the entire dataset is present on the server and no server-client communication is needed. To execute POPri on PubMed, we use the central DP version of POPri, detailed in Algorithm 2. For OpenReview, we employ conditional generation[3] (similar to PE (Lin et al., 2023; Xie et al., 2024), where the generation is conditioned on being given a class to generate. The (central) conditional generation version of POPri is detailed in Algorithm 3.

**Models.** *Next token prediction tasks:* We use LLaMA-3-8B for the LLM $\Psi$ (Grattafiori et al., 2024), which has a knowledge cutoff date of March 2023 (AI@Meta, 2024). For embedding models (used in measuring semantic distance between text samples), we use 'all-MiniLM-L6-v2' sentence transformer (Reimers & Gurevych, 2019b). We

choose DistilGPT2 (Sanh et al., 2019) as the downstream on-device language model for the LargeFedBench evaluations, which has only 82M parameters, and BERT$_{small}$ as the downstream model for the PubMed evaluation to be consistent with Xie et al. (2024). For synthetic text generation (using the LLM $\Psi$), we set the maximum sequence length to 64 for the bioRxiv and Congressional Speeches evaluations and 512 for PubMed/OpenReview.

*Text classification:* We use LLaMA-2-7b-chat-hf for the LLM $\Psi$ (Touvron et al., 2023) to ensure our evaluation was not contaminated, as the knowledge cutoff for LLaMA-2-7b-chat-hf is September 2022 (before the publish date of the ICLR 2023 reviews). For the embedding model we use the 'sentence-t5-xl' sentence transformer (Reimers & Gurevych, 2019a), and use RoBERTa$_{base}$ as the downstream model to be consistent with Xie et al. (2024).

**Metrics.** We primarily evaluate each method on accuracy (next-token or text classification) of the final downstream on-device model $\tilde{\Phi}$. In some ablations we also measure the distance of the synthetic dataset to the private dataset using the Fréchet Inception Distance (FID) (Heusel et al., 2017). During training, we evaluate the models on the validation dataset and select the checkpoint that achieves the best validation performance as the model that is evaluated on the test set.

**Baselines.** We compare POPri to several baselines: (1) DP-FedAvg (McMahan et al., 2017a) (2) DP-FTRL (Kairouz et al., 2021a) (3) Private Evolution (PrE-Text (Hou et al., 2024) and Aug-PE (Xie et al., 2024)). DP-FedAvg and DP-FTRL directly privately fine-tune the downstream model $\Phi$ on the client data. Private Evolution (PrE-Text and Aug-PE) generates synthetic data on which the downstream on-device model $\Phi$ is finetuned. Note that on the PubMed and OpenReview dataset, we compare to Aug-PE results obtained with models of similar size to the model we use (7B-8B parameters) and which are not potentially

---

[3]Simply speaking, we score synthetic data generated for each class separately but combine the preference datasets into one to finetune the LLM $\Psi$.

contaminated (i.e. model was possibly trained on the benchmark dataset). We also include $\epsilon = 0$ (fully private) and $\epsilon = \infty$ (fully non-private) baselines. The $\epsilon = 0$ baseline for the LargeFedBench evaluations evaluates the public Distil-GPT2 checkpoint on the test sets with no further fine-tuning. The $\epsilon = \infty$ baseline is the downstream model finetuned directly on the private training set centralized on the server with no noise. The $\epsilon = 0$ baseline for OpenReview is the accuracy obtained by predicting everything to be the most populous class. More details about the setup can be found in Appendices C and D.2.

**Privacy Analysis.** All baselines use a privacy guarantee of $(\epsilon, \delta)$-DP where $\delta = 3 \times 10^{-6}$ and $\epsilon = 1$ or $\epsilon = 7$ for each of the bioRxiv and Congressional Speeches datasets. For PubMed/OpenReview, we set $\delta < \frac{1}{N_{priv} \cdot \log(N_{priv})}$ ($N_{priv}$ is the number of private samples). We follow the privacy accounting method detailed in Section 3 for POPri. Details for all baselines are in Appendix D.1.

### 5.1. Main Results

Table 1 lists the accuracy (next token prediction and text classification) achieved by baseline methods (DP-FedAvg, DP-FTRL, Private Evolution) and POPri. In this table, we assume full participation (no client sampling) for fair comparison to baselines, some of which do not have client sampling versions. We find that POPri outperforms all the baseline algorithms. Furthermore, in the $\epsilon = 1$ setting POPri closes the gap between fully private learning ($\epsilon = 0$) and fully non-private learning ($\epsilon = \infty$) by 40-58% depending on the setting, compared to PE which closes 1-28%. For all methods tested, the measured accuracy values do not depend strongly on $\epsilon$. This has been observed in prior work on DP synthetic data using LLMs (Xie et al., 2024; Hou et al., 2024). POPri outperforms Private Evolution (Aug-PE) even when holding our synthetic sample budget to 2000. Note that synthetic samples are cheap in POPri (we could generate many more) because we have access to the full model, while Xie et al. (2024) only assume access to a model API.

**Cost analysis case study.** In Table 2 we analyze the per-round communication and computation costs (and per-client, for download/upload/client runtime costs) of FedAvg (a representative and cheap method among the DP-FL-based methods), PE, and POPri on the bioRxiv dataset experiment with 1000 clients sampled per round.

For FedAvg, each round the sampled clients download and upload the downstream model, which in our case is Distil-GPT2. This is an 82M (82 million) parameter model leading to a download and upload cost of 82M floats. The client runtime cost comes from local gradient computation, and server runtime is negligible because the server only needs to average model deltas from the clients. For PE, the

communication cost comes from each client downloading $K = 1800$ sentence embeddings of size 384 resulting in a download cost of 700K (700,000) floats, and uploading a histogram of size 1800 resulting in an upload cost of 1800 floats. The client runtime cost comes from calculating a nearest neighbors histogram and the server runtime cost for PE comes mainly from using the LLM $\Psi$ to generate synthetic samples each round. In POPri each client downloads $K \times J = 1800 \times 10$ sentence embeddings for a download cost of 7M floats and uploads a vector of size 18,000 for an upload cost of 18,000 floats. The client runtime cost of POPri comes from calculating the cosine similarities, and the server runtime comes from both using $\Psi$ to generate synthetic samples and running DPO.

**Interpretation.** In summary, POPri is much more communication-efficient and client compute-efficient than FedAvg, while using much more server compute. On the other hand, POPri is generally more communication- and computationally-expensive than PE. At the same time, POPri has the best downstream performance among all three methods, as seen in Table 1. Hence, POPri can be a suitable method when (1) server compute is cheap and powerful, and (2) getting the best synthetic data/downstream model quality is important.

### 5.2. Ablations

**Cosine similarity vs. Nearest neighbors histogram.** Private Evolution (Lin et al., 2023; Hou et al., 2024; Xie et al., 2024) uses a DP nearest neighbors histogram calculation to score the quality of synthetic samples. The DP nearest neighbors histogram sets the score of a particular synthetic sample to the number of private samples that are closest to that particular synthetic sample (under some text embedding function). In POPri, we instead set the score of a particular synthetic sample to the average cosine similarity between that particular synthetic sample and all private samples (under some text embedding function). We find that cosine similarity works much better than a nearest neighbors histogram (Figure 6), possibly because nearest neighbor histograms produce sparser scores, often assigning zero to all synthetic samples associated with a given prompt. In this setting, the chosen and rejected samples for preference optimization end up being essentially random. In contrast, cosine similarity provides denser scoring that allows the construction of meaningful preference pairs for all prompts.

**Partial client participation.** In each round a fixed number of clients is subsampled uniformly at random for feedback generation. Figure 3 shows the next-token prediction accuracy (%) of four algorithms for different numbers of clients per round. POPri consistently outperforms all of the baselines, regardless of the client sampling rate. Moreover, POPri's accuracy is not sensitive to the client sampling rate.

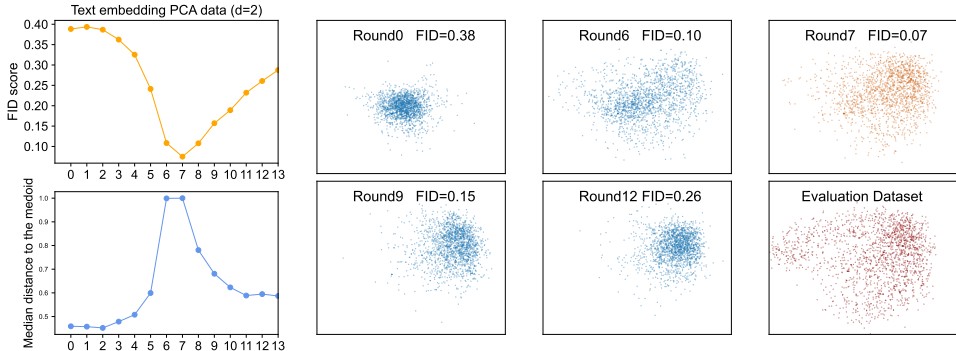

*Figure 4.* PCA visualization of POPri synthetic data embeddings over rounds. **Right 6 Panels:** PCA-2 plots for synthetic data and evaluation data embeddings from the best checkpoint each round for 20 iterations. The orange (round 7) and maroon point clouds represent the round with the lowest FID score and the validation dataset, respectively. **Top Left Panel:** FID score vs. rounds. **Bottom Left Panel:** Median distance to the medoid vs rounds. Running POPri for too many rounds appears to cause overfitting.

| Method | Download (floats) | Upload (floats) | Client Runtime (GPU sec) | Server Runtime (GPU sec) |
|---|---|---|---|---|
| FedAvg | 82 million | 82 million | 4.8 | – |
| PE | 700,000 | 1,800 | 0.0027 | 326.25 |
| POPri | 7 million | 18,000 | 0.01 | 13,547.84 |
| Reduction factor (FedAvg / POPri) | 11.71× | 4555× | 480.0× | – |
| Reduction factor (PE / POPri) | 0.100× | 0.100× | 0.270× | 0.024× |

*Table 2.* **Table setting.** Communication and computation cost comparison per round (and per-client for download/upload/client runtime cost) across methods on the bioRxiv dataset with 1000 clients sampled per round. Download and upload are measured in floats; runtimes are measured in GPU seconds (lower is better). "Reduction factor ($X$ / POPri)" is the cost of method $X$ divided by the cost of POPri for the given resource; green is a reduction, red is an increase. Server runtime for FedAvg is left blank as it is negligible compared to other methods. Overall, we view POPri as suitable when server compute is relatively cheap, and improved sample quality is important enough to justify higher on-device communication and computation costs relative to PE (Table 1).

**Data Distribution Evolution.** Synthetic datasets are often generated using a language model distinct from the one being aligned (Guo et al., 2024), making the alignment phase inherently off-policy as the model evolves during training. This is reflected in the synthetic data, where the FID score (relative to a held-out evaluation set) worsens after improving. Figure 4 shows PCA visualizations of synthetic data embeddings across alignment iterations, while the left panels plot the FID score and median distance to the medoid in the PCA space. The data distribution transitions from being initially clustered to (roughly) matching the true data distribution, back to being clustered, likely due to overfitting. Early stopping based on validation metrics can help.

**How to select rejected samples.** Unlike vanilla DPO, we can select the "chosen" and "rejected" sample pair from the $J$ samples for each of the $K$ prompts. We consistently choose the highest-scoring sample (rank 1) as the "chosen" sample, but there are different options for the "rejected" sample. We found that the middle-ranked sample (e.g., $\ell = 5$th-ranked out of $J = 10$) yields the best results, rather than using the last-ranked sample. If the rejected sample

is too dissimilar to client data, then the preference pair is uninformative. However, choosing a sample that is too similar to client data (e.g., rank 2) for the rejected sample could lead to incorrect preference pairs due to DP noise swapping rankings. We use the 5th-ranked sample, and justify it experimentally in Appendix E.3.

## 6. Conclusion

Private on-device learning is important when data is stored on edge devices with hardware, storage, and privacy constraints. We propose POPri, which recasts synthetic data-based approaches for private learning as an LLM policy optimization problem. POPri makes several novel design choices in how it gathers and utilizes client feedback to generate DP synthetic data, which is used to finetune a downstream on-device model. POPri outperforms DP-FL and synthetic data baselines on a variety of tasks, including on a large-scale LargeFedBench, a new federated benchmark we have curated.

## Impact Statement

In this paper, we train models satisfying differential privacy guarantees. When using differential privacy as a tool for protecting user data, it is important to communicate to users what the privacy guarantees mean to be able to obtain informed consent. The algorithms in this paper also use LLMs, which were trained on large scale public text data. While this data was public, explicit consent may not have been given for its use in training the models. The algorithms using LLMs in the paper make no claims about the privacy guarantees of data used in the pretraining of the LLMs.

While our work aims to show how synthetic data can be useful for federated learning, it also poses a number of ethical risks, including the generation of biased or harmful content. In particular, our method (and all variants of private evolution) inherits the biases and undesirable aspects of the public LLM. For example, suppose the public LLM only generates text in English, but some clients' private data is all in Spanish. In these settings, clients would be forced to vote on synthetic samples, even if potentially none of them are relevant to the client. This may cause the client to contribute data reinforcing a model that is actively not useful (or even harmful) to the client. In contrast, DP-SGD methods do not suffer from this shortcoming, because they do not rely on a public LLM. This problem raises an important point—how can we design DP synthetic data algorithms in which clients can stem the biases or failures of the public LLM, based on their own data? This important question is beyond the scope of the current paper.

## Acknowledgments

This work was supported in part by NSF grants CCF-2338772 and CNS-2148359, as well as C3.ai, Bosch, Intel, and the Sloan Foundation. This work used Bridges-2 GPU (Brown et al., 2021; Buitrago & Nystrom, 2021) at the Pittsburgh Supercomputing Center through allocation CIS240135 and CIS240937 from the Advanced Cyberinfrastructure Coordination Ecosystem: Services & Support (ACCESS) program, which is supported by National Science Foundation grants 2138259, 2138286, 2138307, 2137603, and 2138296 (Boerner et al., 2023). The authors acknowledge the National Artificial Intelligence Research Resource (NAIRR) Pilot, the AI and Big Data group at the Pittsburgh Supercomputing Center, and NCSA Delta GPU for contributing to this research result.

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

# A. Algorithmic Details

Here we provide the pseudocode for the centralized version of POPri and the central + conditional generation version of POPri.

---

**Algorithm 2** `POPri` (central DP, unconditional)

---

1: **Input:** Number of iterations $T$, Noise multiplier $\sigma$, LLM $\Psi$, embedding model $\Gamma$, base prompt $\eta$, random prompt generator $\Lambda(\cdot)$, "rejected" index $\ell$, private dataset $S$, $K$ number of prompts, $J$ number of responses per prompt
2: **Output:** LLM for generating synthetic data $\Psi_{T+1}$
3:
4: Embed all private samples $E = \Gamma(S)$
5: Initialize LLM $\Psi_1 = \Psi$
6: **for** $t \leftarrow 1 \ldots T$ **do**
7:     Initialize the response vector $R = \emptyset$
8:     **for** $k \leftarrow 1 \ldots K$ **do**
9:         Generate prompt $\eta_k = \Lambda(\eta)$,
10:         Generate $J$ responses $R_{kj} = \Psi_t(\eta_k), j \in [J]$
11:     **end for**
12:     Calculate embeddings $E_{syn,t} = \{\Gamma(R_{kj})\}_{k \in [K], j \in [J]}$
13:     $\text{Scores}_t \leftarrow \texttt{CENTRALSCORE}(E_{syn,t}, E) + \mathcal{N}(0, \sigma^2 I)$
14:     Set $P[k, j]$ as the $j$-th highest score response for prompt $\eta_k$, according to $\text{Scores}_t$
15:     Initialize preference dataset $\mathcal{P}_t = \emptyset$
16:     **for** $k \leftarrow 1 \ldots K$ **do**
17:         Select positive synthetic sample: $\mathcal{P}_t[k, 1] = P_t[k, 1]$
18:         Select negative synthetic sample: $\mathcal{P}_t[k, 2] = P_t[k, \ell]$
19:     **end for**
20:     Fine-tune: $\Psi_{t+1} \leftarrow \text{DPO}(\Psi_t, \{\eta_k\}_{k \in [K]}, \mathcal{P}_t)$
21: **end for**
22: Output $\Psi_{T+1}$

---

**Algorithm 3** `POPri` (central DP, conditional)

---

1: **Input:** Number of iterations $T$, Noise multiplier $\sigma$, LLM $\Psi$, embedding model $\Gamma$, base prompt $\eta$, conditional (class-specified) random prompt generator $\Lambda(\cdot, \cdot)$, "rejected" index $\ell$, private dataset $S$, $K$ number of prompts, $J$ number of responses per prompt, number of classes $B$
2: **Output:** LLM for generating synthetic data $\Psi_{T+1}$
3: Embed private samples for each class $i = 1 \ldots B$, $E_i = \Gamma(\{s\}_{F(s)=i, s \in S})$ where $F(s)$ is the class index of sample s
4: Initialize LLM $\Psi_1 = \Psi$
5: **for** $t \leftarrow 1 \ldots T$ **do**
6:     Initialize B response vectors $R = \{\emptyset, \ldots \emptyset\} = \{R^{(1)}, \ldots, R^{(B)}\}$
7:     **for** $b \leftarrow \ldots B$ **do**
8:         **for** $k \leftarrow 1 \ldots K$ **do**
9:             Generate prompt $\eta_k = \Lambda(\eta, b)$,
10:             Generate $J$ responses $R_{kj}^{(b)} = \Psi_t(\eta_k), j \in [J]$
11:         **end for**
12:         Calculate embeddings $E_{syn,t}^{(b)} = \{\Gamma(R_{kj}^{(b)})\}_{k \in [K], j \in [J]}$
13:         $\text{Scores}_t \leftarrow \texttt{CENTRALSCORE}(E_{syn,t}^{(b)}, E_b) + \mathcal{N}(0, \sigma^2 I)$
14:         Set $P[k, j]$ as the $j$-th highest score response for prompt $\eta_k$, according to $\text{Scores}_t$
15:         Initialize preference dataset $\mathcal{P}_t^{(b)} = \emptyset$
16:         **for** $k \leftarrow 1 \ldots K$ **do**
17:             Select positive synthetic sample: $\mathcal{P}_t^{(b)}[k, 1] = P_t[k, 1]$
18:             Select negative synthetic sample: $\mathcal{P}_t^{(b)}[k, 2] = P_t[k, \ell]$
19:             Set prompt $\mathcal{P}_t^{(b)}[k, 3] = \eta_k$
20:         **end for**
21:     **end for**
22:     Fine-tune: $\mathcal{P}_t = \bigcup_{b=1}^{B} P_t^{(b)}$, $\Psi_{t+1} \leftarrow \text{DPO}(\Psi_t, \mathcal{P}_t)$
23: **end for**
24: Output $\Psi_{T+1}$

---

Below is the similarity scoring function we use for the federated setting.

---

**Algorithm 4** `SIMILARITY`

---

1: **Input:** Set of embeddings of private client data $E_i = \{emb(s_1^{(i)}), \ldots, emb(s_{m_i}^{(i)})\}$ for $i \in \mathcal{S}^t$, embeddings of synthetic data $E_{syn}$, total synthetic samples $M = K \times J$
   Scores $\leftarrow \mathbf{0}^M$
2: Scores$[j] = \frac{1}{m_i} \sum_{e_{pri} \in E_i} \frac{\langle e_{pri}, e_j \rangle}{\|e_{pri}\|\|e_j\|}$ for $e_j \in E_{syn}$
3: **return** Scores$/\max(1, \|\text{Scores}\|_2)$

---

Below is the similarity scoring function we use for the central DP setting.

---

**Algorithm 5** `CENTRALSCORE`

---

1: **Input:** Embeddings of private data $E$, embeddings of synthetic data $E_{syn}$
   Scores $\leftarrow \mathbf{0}^M$
2: Scores$[j] = (1/|E|) \sum_{e_{pri} \in E} \frac{\langle e_{pri}, e_j \rangle}{\|e_{pri}\|\|e_j\|}$ for $e_j \in E_{syn}$
3: **return** Scores

---

# B. Implementation Details of POPri

## B.1. Model and Hyperparameters

We choose LLaMA-3-8B as the data generator in POPri and we fine-tune it iteratively during the course of the algorithm. To fine-tune the LLaMA-3-8B model, we use LoRA fine-tuning with rank 4, $\alpha = 8$, applied to all the projection matrices in LLaMA-3-8B. We adapt the AdamW optimizer with a cosine learning rate scheduler with the learning rate ranging from $3 \cdot 10^{-7}$ to $8 \cdot 10^{-7}$. In the Congress and bioRxiv evaluations, the sample set $\Omega$ is a subset of the c4 dataset (Raffel et al., 2019), which is a large scale dataset from 2019, which we use for fair comparison with Private Evolution (PrE-Text), though we do not know their exact initial sample set because they did not release it. For the PubMed evaluation, the sample set $\Omega$ is a set of 2000 samples generated using the PubMed generation prompt in Table 16 of the Aug-PE paper, generated by LLaMA-3-8B-Instruct (which has a knowledge cutoff of March 2023), for comparison with Aug-PE (Xie et al., 2024). For each iteration, we fine-tune the models for 2 epochs and select the best checkpoint with the lowest FID score relative to the validation dataset. This checkpoint is used for synthetic data generation and as the starting point for the next iteration. The batch size is set to 24.

In each round we generate 18000 synthetic data samples for the clients to evaluate. This is accomplished with 1800 prompts, each generating 10 samples for clients to rank. We select the 1st and 5th ranked sample for a given prompt for the "selected" and "rejected" data samples in the DPO preference dataset. We describe the experiments regarding which rank to use for constructing the preference dataset in detail in Appendix Section E.3. To test the scaling relation with the number of clients per round and the total number of clients participating in the training, we set up the parameters and privacy budget shown in Table 3. The 'all-MiniLM-L6-v2' sentence transformer model is used as the embedding model in POPri. We note that we adopt "sentence-t5-base" sentence transformer for PubMed during the step of fine-tuning $\text{BERT}_{small}$, which follows the setting in AUG-PE. We ensure POPri follows privacy guarantee of $(\epsilon, \delta)$-DP $= (1, 3 \times 10^{-6})$ or $(7, 3 \times 10^{-6})$ for both the bioRxiv and the Congressional Speeches datasets and run with 20 iterations for DP-FedAvg, DP-FTRL, PrE-Text for comparison. For AUG-PE, we set $(\epsilon, \delta)$-DP $= (1, 2.72 \times 10^{-6})$ or $(4, 2.72 \times 10^{-6})$. PubMed experiments are run with 10 iterations.

In terms of models for downstream tasks:

- For BioRxiv & Congressional Speeches, we fine-tuned the pre-trained DistillGPT2 for next-token prediction. We set the max sequence length as 64, number of generated synthetic data as 1,000,000, the batch size as 160, the learning rate as $2e^{-4}$, and the number of epochs as 80.

- For PubMed, to compare with (Yue et al., 2023b), we follow their procedure to leverage pre-trained $\text{BERT}_{small}$ (Turc et al., 2019). We set the max sequence length as 512, number of generated synthetic data as 2000, batch size as 32, learning rate as 3e-4, the weight decay as 0.01, and the number of epochs as 10. To compare with (Xie et al., 2024), we set up the $(\epsilon, \delta)$-DP value and hypterparameter according to their choice. For example, they set $\delta = \frac{1}{N_{priv} \cdot \log(N_{priv})}$ following (Yue et al., 2023b). To achieve $\delta = \{1, 4\}$, we use noise multiplier $\sigma = \{13.7, 3.87\}$ for 10 iterations under DP

List of 6 diverse original text samples:

Original Text Sample 1
The observations showed that the object is four million times more massive than the sun and is the size of one astronomical unit (AU), a span equal to Earth's distance from the sun. Sgr A* has a mass density at least a trillion times greater than any known cosmic object.

Original Text Sample 2
In response to the general question, they need to study self-protection away from their marital baggage. They need to learn about home security, mobile security, the nature of crime, de-escalation, the law, escape tactics, awareness, and on and on. When it

Original Text Sample 3
Under the Patriot Act of 2001, the government significantly expanded its authority in regards to electronic surveillance (Henderson, 2002). One of the chief complaints is that the government can investigate anything that is considered "significant." The problem here is that there is

Original Text Sample 4
The life history advance program shall be funded from any of the following: monies provided by the general fund; amounts in the presidential family partnership fund; or monies provided by the revolving fund.

Original Text Sample 5
As you meet with employers this summer, get in touch with the team….

*Figure 5.* The synthetic data generation prompt for POPri. The black text marks the input prompt, and the brown text after "Original Text Sample 4" is generated. The generated text between "Original Text Sample 4" and "Original Text Sample 5" is collected and used as a synthetic sample.

on all PubMed data. Note that our noise multiplier values are slightly different than (Xie et al., 2024) due to different methods for calculating differential privacy.

### B.2. Prompt Design

To compare with other data generator methods, we adopt the prompts used in the baseline models against which we compare. We generate the synthetic data using an approach similar to that in PrE-Text (Hou et al., 2024). Figure 5 shows an example of the prompt we use for prompting LLaMA-3-BB for generating synthetic data. For bioRxiv/Congress, we randomly take text samples from the c4 (Raffel et al., 2019) dataset as our examples in the prompt. For PubMed, while running POPri, we still adopt the prompt shown in Figure 5 but reduce the number of examples to two in order to accommodate longer sequence lengths, randomly sampling generated abstracts from LLaMA-3-8B. For OpenReview, we prompt the model directly to generate paper reviews (similarly to (Xie et al., 2024)).

## C. Implementation Details of Baseline Models

In this section we provide implementation details for the baseline algorithms. We use two DP-FL baselines: DP-FedAvg and DP-FTRL. For the PE baseline, we implement PrE-Text (Hou et al., 2024) for the evaluations on the bioRxiv and Congressional Speeches datasets. For the PE baselines on the PubMed dataset we directly compare against the Aug-PE results from Xie et al. (2024).

### C.1. DP-FedAvg

We employ the FedAvg federated optimization algorithm (McMahan et al., 2017b) to fully fine-tune DistilGPT2, avoiding linear probing due to its poor performance in DP language models (Lin et al., 2021). Our training configuration includes a batch size of 2, a sequence length of 64, and 20 rounds for Table 1 and 50 rounds for Figure 3, and either full or partial client participation. For differential privacy (DP), we utilize secure aggregation (Bonawitz et al., 2016) and introduce Gaussian noise (McMahan et al., 2017b). We evaluate the model using next-token prediction accuracy across various numbers of training epochs on the clients. We tune the learning rate within the range [0.001, 0.1, 0.1] and the clipping threshold between [0.01, 0.1, 1.0], selecting the model with the best performance on the evaluation set for reporting. The noise is scaled to ensure a privacy guarantee of $(\epsilon, \delta)$-DP where $\delta = 3 \cdot 10^{-6}$ and $\epsilon = \{1, 7\}$, representing two distinct privacy regimes. The noise multipliers are $\sigma = \{19.3, 3.35\}$ when considering all the data, and the settings for partial participation experiments

*Table 3.* Experiment privacy budget settings.

| Total # of clients | # of clients per round | $\sigma_1{}^a, \epsilon = 7$ | $\sigma_1{}^a, \epsilon = 1$ | $\sigma_2{}^b, \epsilon = 7$ | $\sigma_2{}^b, \epsilon = 1$ |
|---|---|---|---|---|---|
| 10000 | 1000 | — | 3.4 | — | 19.5 |
| 10000 | 5000 | — | 15.5 | — | 30.8 |
| 10000 | 10000 | — | 30.6 | — | 30.8 |
| 72000 | 72000 | 3.35 | 19.3 | 3.35 | 19.5 |
| 133000 | 133000 | 3.35 | 19.3 | 3.35 | 19.5 |

[a] For DP-FedAvg, PrE-Text, POPri.
[b] For DP-FTRL

are shown in Table 3.

## C.2. DP-FTRL

We also use the DP variant of Follow-The-Regularized-Leader (DP-FTRL) algorithm (Kairouz et al., 2021a) to fully fine-tune DistilGPT2. The hyperparameter settings are similar to DP-FedAvg other than the noise multipliers. The noise multipliers are $\sigma = \{19.5, 3.35\}$ when considering all the data, and the settings for partial participation experiments are shown in Table 3.

## C.3. PrE-Text

We follow similar settings as Hou et al. (2024) with some modifications. The privacy budget is similar to DP-FedAvg and POPri, with a privacy guarantee of $(\epsilon, \delta)$-DP where $\delta = 3 \cdot 10^{-6}$ and $\epsilon = \{1,7\}$ with $\sigma = \{19.3, 3.35\}$ for full participation and partial participation in Table 3. We set the thresholds H = 0.1626, T = 20, and $N_{syn} = 1024$. We adopt the "all-MiniLM-L6-v2" sentence transformer model for text embedding generation.

# D. Experimental Details

## D.1. Privacy Accounting

The precise privacy settings we use and their corresponding $\epsilon$ values, as calculated by their corresponding privacy budget computation methods, are reported in Table 3. DP-FedAvg (McMahan et al., 2017b) and Private Evolution (PrE-Text) (Hou et al., 2024) both use the Gaussian mechanism, and thus use similar computations. In both cases, we use the privacy accountant of the Opacus library (Yousefpour et al., 2021). For DP-FedAvg, we calculate privacy by inputting the number of rounds, the client sampling ratio, setting the noise multiplier to be the product of $\sigma$ and the clipping threshold, choosing a $\delta \ll 1/|\mathcal{S}|$, and setting $\sigma$ for the desired $\epsilon$. Private Evolution (PrE-Text) (Hou et al., 2024) also uses the Gaussian mechanism, so we use the same accounting except the noise multiplier is the product of $\sigma$ and the maximum number of samples per client. For DP-FTRL, we follow the privacy accounting methods from their implementation. For Private Evolution (Aug-PE) (Xie et al., 2024), we report their reported $\epsilon$ directly.

## D.2. Evaluation Details for Different Datasets

### D.2.1. LARGEFEDBENCH EVALUATION

For the bioRxiv and Congressional Speeches datasets, we use the PrE-Text version of Private Evolution because the PrE-Text evaluation focused on datasets with samples with max sequence length of 64.

### D.2.2. PUBMED AND OPENREVIEW EVALUATION

For PubMed and OpenReview, our Private Evolution baseline compares to Aug-PE, which has already been evaluated on PubMed and OpenReview (Xie et al., 2024). Note that PubMed and OpenReview was used by Xie et al. (2024) to evaluate central DP algorithms. In the central DP setting, there are no clients; all private data is held at the server and the goal is to release a model with DP guarantees. The notion of neighboring dataset in central DP is a centrally held dataset that is the same except for a single data sample. To compare our algorithm directly with results reported for Private Evolution (Aug-PE) (Xie et al., 2024), we replicate the central DP setting for this dataset by having one PubMed abstract per client and

sampling all clients every iteration (or "round", in our case).

# E. Ablation Studies

## E.1. Cosine similarity vs Nearest neighbors histogram

In this section we perform an ablation justifying the choice of cosine similarity as a scoring function over the nearest neighbor histogram employed by Private Evolution. We find that using cosine similarity works much better than nearest neighbors histogram for our use case, because nearest neighbors histogram is too sparse to ensure the construction of meaningful preference pairs for POPri.

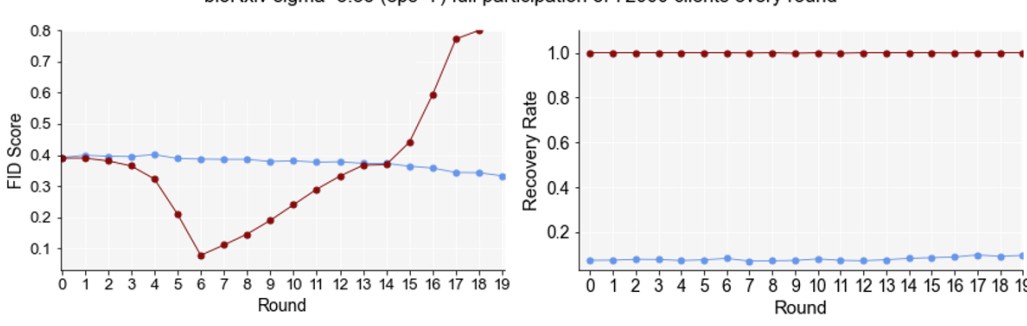

*Figure 6.* **Left:** FID scores of POPri using NN histogram scoring vs. POPri using cosine similarity. **Right:** After the client feedback stage, we measure the percentage of the time the non-noised and non-clipped score (nearest neighbor histogram scoring or cosine similarity scoring) of the chosen sample is higher than the rejected sample. For cosine similarity, this "recovery rate" is much higher (nearly 100%) than in nearest neighbors histogram. **Interpretation.** Nearest neighbors histogram is much sparser than cosine similarity, often assigning zero to all synthetic samples associated with a given prompt in POPri. This leads to preference pairs often being completely noisy. Cosine similarity provides denser scoring that allows the construction of meaningful preference pairs for all prompts.

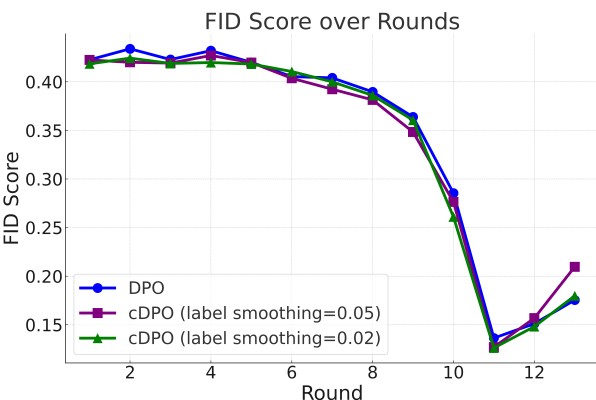

*Figure 7.* In this experiment, we investigate whether the use of label noise-resistant alignment methods could allow the use of higher-ranked rejected samples. To do this, we used the third-ranked sample as the rejected sample, and evaluated different settings for conservative DPO (cDPO) (Mitchell, 2023). We used the bioRxiv dataset experiment setting, set eps=7, learning rate = 8e-7. We find that by tuning the level of conservative-ness we may be able to improve slightly on vanilla DPO.

## E.2. Alignment methods

We also experiment with a noise-resistant (or robust) DPO method, conservative DPO (cDPO) (Mitchell, 2023), to see if by using it we can select a higher ranked rejected sample (recall we use the 5th ranked, and higher ranked samples would introduce more noise into the preference pairs). In Figure 7 we find that it can help slightly when choosing a higher rejected sample ranking.

### E.3. Rejected sample selection

We construct the DPO preference data via client feedback by generating ten samples from the same prompt and then picking the "selected" and the "rejected" samples. The samples with the highest scores among the ten examples are picked as the "selected" sample in the DPO preference dataset. We experiment on which rank should be utilized as the "rejected" sample in the DPO preference dataset. In Fig 8 we further explore the effects by examining the "rejected" and "selected" sample FID scores as a function of round. In the left panel where the "selected" sample FID values are shown, their magnitude and trends behave similarly before they reach the best results (marked by colored dashed vertical lines). For the "rejected" sample FID shown in the right panel, the 5th rank "rejected" samples yield the lowest FID score and therefore smaller gap between the preference sample pairs. However, we also find that higher rank does not always yield better results. This may result from the boundary between the "rejected" and "selected" samples becoming undistinguishable for rank < 5th due to DP noise. We therefore select 5th rank samples as our "rejected" DPO preference samples.

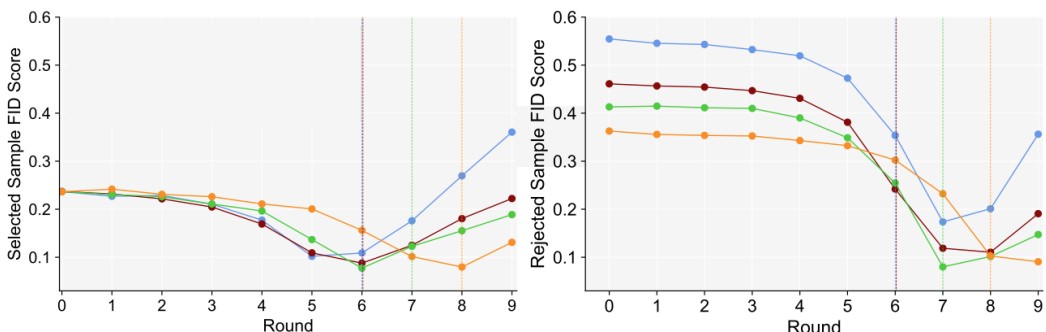

*Figure 8.* Ablation study for selecting rejected samples in the preference data. Here we generate 10 samples for each prompt and select Nth ranked data as the rejected sample, where N is 5, 7, or 10. The vertical lines indicate the round at which the best next-word-prediction accuracy was achieved for each choice of rank. Note that the model that produces the lowest overall FID (not the lowest selected sample FID or the lowest rejected sample FID) is the best synthetic data generation model, since on the final round all generated samples are utilized to form the synthetic dataset. We hypothesize that round 7 corresponds to the highest accuracy for the rank 5 model because after that point, the selected sample FID is higher than the rejected sample FID, which would mean the preference dataset has become mis-aligned with the objective of generating good synthetic data.

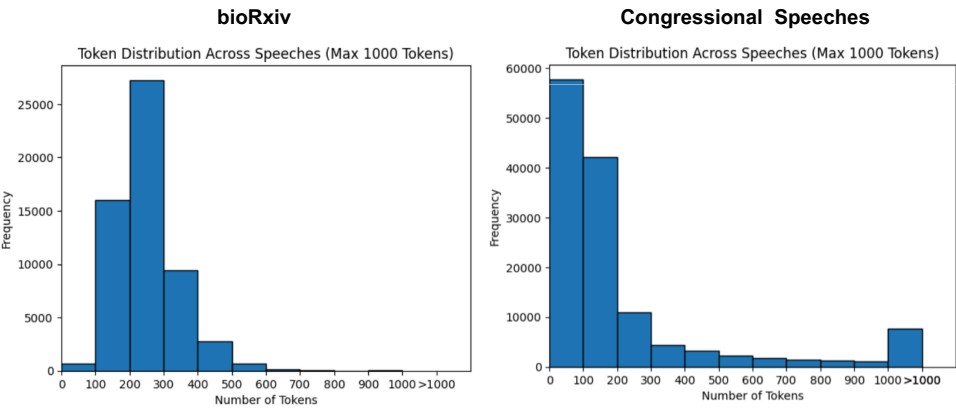

*Figure 9.* The distribution of how many tokens are in each client's dataset for the bioRxiv and Congressional Speeches datasets.

*Table 4.* Dataset details.

| Dataset | # Train Samples | # Validation Samples | # Test Samples | Max Sequence Length | Average # of samples per client |
|---|---|---|---|---|---|
| bioRxiv | 72000 | 2000 | 1584 | 64 | $6.6 \pm 2.6$ |
| Congressional Speeches | 133000 | 4200 | 1547 | 64 | $5.0 \pm 16.3$ |
| PubMed | 75316 | 14423 | 4453 | 512 | 1 |

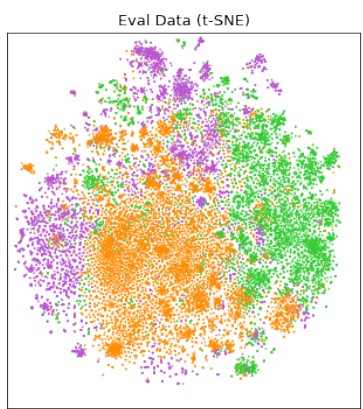

*Figure 10.* A t-SNE clustering of the Congressional Speeches dataset. US data is colored in purple, UK data is colored in orange, and Canada data is colored in green. We find that the three datasets form distinct clusters and also distinct sub-clusters.

## F. Datasets

**bioRxiv.** This dataset consists of abstracts from bioRxiv papers with appropriate copyright permission from April 2023 to August 2024. This was done by using the bioRxiv public API to retrieve the abstracts of the paper with permitted licenses (i.e. 'CC BY NC ND', 'CC BY ND', 'CC BY NC', 'CC BY', 'CC0'). This dataset consists of 72k abstracts (clients), each of which we split into chunks of 64 tokens to form samples.

**Congressional Speeches.** This dataset consists of speeches from US, UK and Canada congressional/parliamentary transcripts from April 2023 to August 2024. All speeches are published under a permissive license which allows for third-party use (as detailed in the dataset cards). There are 134k speeches (clients) in total, and 1930 unique speakers. We collected this dataset by using public APIs to retrieve data from each country's official congressional/parliamentary library website. Then we sanitized the data by removing (1) boilerplate procedural language, (2) sentences with more than 30% of the characters not being letters, and (3) some written notation that does not correspond to spoken words. We split each speech into chunks of 64 tokens each. We believe that this dataset is a major contribution because spoken language may be more resistant to contamination (especially for the UK and Canada parliamentary debates). Because they are more conversational and have a large degree of improvisation (many debates are off-the-cuff), they are less likely to be generated by LLMs. Because Congressional Speeches contains a diverse collection of speeches across speakers and also countries, the dataset forms many distinct clusters, reflecting the diversity of the dataset (Figure 10).

We will update the dataset periodically with the latest data to allow future researchers to test their algorithms or ideas against an uncontaminated dataset.

