# OpenReview forum: "Private Federated Learning using Preference-Optimized Synthetic Data"
_ICML.cc/2025/Conference — ICML 2025 poster_

### Official Review · Reviewer_W8ui · 2025-03-13

**Overall Recommendation:** 2

**Summary:**

The paper introduces ​POPri, a novel method for private on-device learning that leverages DP synthetic data generated via LLMs. They use ​Direct Preference Optimization (DPO) to fine-tune LLMs for generating high-quality synthetic data. POPri outperforms existing methods, in terms of next-token prediction accuracy, particularly on the newly introduced ​LargeFedBench benchmark.

**Claims And Evidence:**

See weaknesses and questions.

**Essential References Not Discussed:**

NA

**Ethical Review Concerns:**

This submission significantly changed the original ICML template.

**Ethical Review Flag:**

Flag this paper for an ethics review.

**Ethics Expertise Needed:**

["Other expertise"]

**Experimental Designs Or Analyses:**

See weaknesses and questions.

**Methods And Evaluation Criteria:**

See weaknesses and questions.

**Other Comments Or Suggestions:**

Please do not change the template format.

**Other Strengths And Weaknesses:**

**Strengths**

1.  The authors introduce ​LargeFedBench, a new federated benchmark for LLM evaluation, which includes ​Congressional Speeches and ​bioRxiv datasets. This benchmark is designed to avoid contamination and provides a large number of clients, making it a valuable resource for future research.

2. The paper introduces a novel approach to private on-device learning by framing it as an LLM preference optimization problem. This is a significant departure from traditional DP-FL and synthetic data methods, which rely on prompt engineering or in-context learning.

**Weakness**

1. It presents very limited downstream task evaluations. We can hardly judge if it works for a broader scope of downstream tasks.

2.  The experiments are primarily focused on text data (e.g., bioRxiv, Congressional Speeches). It would be beneficial to see how POPri performs on other types of data, such as images or structured data, to assess its generalizability.

3. This paper uses DPO to improve synthetic data generation, but it does not compare it with other possible methods, such as RLHF.

4. While the paper briefly mentions the ethical implications of using LLMs trained on public data, it does not delve deeply into the potential risks of synthetic data generation, such as the possibility of generating biased or harmful content.

**Questions For Authors:**

1. The paper uses DPO to fine-tune the LLM. Why was DPO chosen over other preference optimization methods like RLHF? What are the specific advantages of DPO in the context of generating DP synthetic data?

2. The paper claims that POPri has significant communication and computation cost advantages over DP-FL. Can you break down the specific cost savings in terms of client upload/download communication and computation, and how these savings are achieved?

3. In the context of POPri, how does the choice of the rejected sample (e.g., middle-ranked vs. last-ranked) impact the quality of the synthetic data generated? What is the rationale behind selecting the middle-ranked sample as the rejected sample?

**Relation To Broader Scientific Literature:**

NA

**Theoretical Claims:**

No theoretical claims.

---

> ### Author Rebuttal · Authors · 2025-04-01
>
> **It presents very limited downstream task evaluations…It would be beneficial to see how POPri performs on other types of data**
>
> On more tasks: Thank you for this comment—we are working on an evaluation of a text classification task. Due to the tight deadline of the rebuttal preparation period, we will provide the results in the coming days as the results finish running.
>
> On other modalities: We completely agree that our evaluation does not allow one to generalize to other modalities. Focusing on a single modality is common in recent DP synthetic data papers, including several works that have won oral or spotlight awards at recent top ML conferences [1-5]. We will clarify in the introduction and abstract that our paper covers only the text modality.
>
> [1] Lin et al, “Differentially private synthetic data via foundation model APIs 1: Images”, ICLR 2023
> [2] Xie, C. et al, Differentially private synthetic data via foundation model APIs 2: Text, ICML 2024  (Spotlight)
> [3] Hou, C. et al, Pre-text: Training language models on private federated data in the age
> of LLMs, ICML 2024 (Oral)
> [4] Qian et al. "Synthcity: a benchmark framework for diverse use cases of tabular synthetic data." NeurIPS 2023
> [5] Dockhorn et al. "Differentially Private Diffusion Models." Transactions on Machine Learning Research 2022.
> [6] Lin et al, "Differentially Private Synthetic Data via APIs 3: Using Simulators Instead of Foundation Model." 2025
>
>
> **Why was DPO chosen over other preference optimization methods like RLHF?**
>
> Thank you for your comment. We compared against IPO (Identity Preference Optimization) [7] in our paper which is a common mainstream alternative method to DPO in Table 3 of the appendix. We found that IPO did not perform as well as DPO.
>
> [7] A General Theoretical Paradigm to Understand Learning from Human Preferences
> Gheshlaghi Azar et al., 2023, https://arxiv.org/abs/2310.12036
>
> **While the paper briefly mentions the ethical implications of using LLMs trained on public data, it does not delve deeply into the potential risks of synthetic data generation, such as the possibility of generating biased or harmful content.**
> Thank you for noting this—we will expand our Impact Statement to discuss each of these important points.
>
> **Can you break down the specific cost savings in terms of client upload/download communication and computation, and how these savings are achieved?**
>
> Thank you for the comment–our description of the communication and computation cost savings was not clear. First, we describe the communication savings. In our experiment setting, DP-FL clients have to upload and download a DistilGPT2 model, which consists of 82M floats. Meanwhile, POPri communicates K x J = 1800 x 10 = 18000 text embeddings of size 384 for a total of 7M floats to download. POPri clients communicate back a vector of size 18000 (one score per text) to the server, resulting in 18K floats to upload. We have
>
>
> | Method   | Download (floats) | Upload (floats) |
> |--|--|--|
> | DP-FL    | 82M        | 82M       |
> | POPri    | 7M        | 18,000           |
> | Reduction Factor (DP-FL / POPri) | **~11.7×**         | **~4555.6×**        |
>
> Next, we provide a cost savings calculation in terms of compute. Experiment setting: we sample 1000 clients per round on the bioRxiv dataset:
>
> | Method   | Client Runtime (V100 GPU) in GPU seconds | Server Runtime (A100 GPU) in GPU seconds     |
> |--|--|---|
> | DP-FL   | 4.8 sec                    | -                               |
> | POPri    | 0.01 sec                   | 13,547.84 sec |
>
> The client compute cost is small for POPri (about 500x smaller) than DP-FL. Meanwhile, the server side compute cost is small for FedAvg, and large for POPri. POPri is suitable in cases where (1) we care a lot about the downstream model quality, (2) the client computation and/or communication is expensive, and (3) server computation is relatively cheap and powerful. We will include this analysis in the paper.
>
> **In the context of POPri, how does the choice of the rejected sample (e.g., middle-ranked vs. last-ranked) impact the quality of the synthetic data generated? What is the rationale behind selecting the middle-ranked sample as the rejected sample?**
>
> This choice was a heuristic based on our experiments, discussed partially in Appendix E.4. Choosing very low-ranked samples (namely, rank 10) was bad–during preference optimization, the model was learning to distinguish between a good sample and a very bad sample, which distorted its performance. We needed to select more challenging preference pairs for DPO. However, if we had chosen rank 3 as the negative sample (for instance), the DP noise would be more likely to swap the ordering of the first and second-ranked samples. Hence, as an empirically-motivated compromise, we chose the 5th-ranked sample, which performed the best in our ablation.

---

### Official Review · Reviewer_xvTm · 2025-03-13

**Overall Recommendation:** 3

**Summary:**

The authors introduce a client-level differentially private (DP) federated learning algorithm that leverages synthetic data generation assisted by large language models (LLMs). Unlike previous approaches that rely only on prompting for synthetic data generation, their proposed POPri algorithm fine-tunes the LLM's weights using Direct Preference Optimization (DPO). Through extensive experiments, they demonstrate the effectiveness of POPri, showing improved performance compared to existing methods.

**Claims And Evidence:**

Some of the claims are sort of confusing to me, especially on the part of privacy guarantees. In particular, I have the following questions:
1. The threat model is unclear to me. Are the authors considering trusted server or un-trusted server model? Also, why do the authors add noise both on client and server sides (as in line 17 and 20 in Algorithm 1)?
2. The DP nearest neighbors algorithm shown in Algorithm 2, is it a differential private algorithm since I don't see any privatization technique used in Algorithm 2?
3.  what do $e_{pri}$ and $e_j$ represent in Algorithm 2? Is $E_i$ just one embedding or a collection of embeddings?

**Essential References Not Discussed:**

N/A

**Experimental Designs Or Analyses:**

The experimental design makes sense to me. However, the performance improvement is not particularly surprising, given that the proposed method is significantly more resource-intensive than the baselines. Compared to DP-FedAvg and DP-FTRL, POPri benefits from LLM-assisted synthetic data generation. Meanwhile, compared to PE, POPri chooses to fine-tune the LLMs instead of prompting, which is more computationally demanding.

That said, the paper does not clearly address the cost-utility tradeoff of this approach. While it covers communication and computation complexity on the client side, most of the computation occurs on the server side. I believe it is essential to include a discussion on the overall computational complexity of the proposed method (both client and server sides) relative to existing approaches.

**Methods And Evaluation Criteria:**

The proposed methods seem reasonable to me. However, the approach appears to be a straightforward extension of private evolution algorithms where POPri utilizes privatized clients responses to fine-tune the LLMs responsible for data generation. This may limit the paper's overall contribution. Additionally, I am concerned about the computational complexity of this method, as fine-tuning is likely to be significantly more resource-intensive than prompting.

**Other Comments Or Suggestions:**

N/A

**Other Strengths And Weaknesses:**

N/A

**Questions For Authors:**

In Line 20 of Algorithm 1, why does the server aggregate the scores from all clients instead of the clients in $\mathcal{S}^t$?

**Relation To Broader Scientific Literature:**

N/A

**Theoretical Claims:**

There are no theoretical claims in the paper but I suggest the authors giving a clear argument on the privacy analysis.

---

> ### Author Rebuttal · Authors · 2025-04-01
>
> **Are the authors considering trusted server or un-trusted server model? Also, why do the authors add noise both on client and server sides (as in line 17 and 20 in Algorithm 1)?**
>
> Thank you for pointing out the typo; we should only add noise in line 17. We consider the server to be untrusted. Our reported (epsilon, delta) are with respect to an adversary that can see only aggregated updates at every round during training, similar to past work on FL[1,2]. We will clarify.
>
> [1] Kairouz et al, 2021 “Practical and Private (Deep) Learning without Sampling or Shuffling” https://arxiv.org/abs/2103.00039
>
> [2] McMahan et al, 2017 “Learning Differentially Private Recurrent Language Models” https://arxiv.org/abs/1710.06963
>
> **I don't see any privatization technique used in Algorithm 2? what do e_pri and e_j represent in Algorithm 2? Is E_i just one embedding or a collection of embeddings?**
>
> First, we would like to apologize for an oversight; Algorithm 2 is not the one actually used in our experiments. In POPri the clients actually score each synthetic sample based on the average cosine similarity between the synthetic embedding and their private data embeddings. The experimental results reported in the paper use this cosine similarity scoring method. Updated pseudocode for Algorithm 2 is below:
>
> >
> > 1. **Input:** Embeddings of private client data $E_i$ for $i \in S^t$, embeddings of synthetic data $E_{syn}$, total synthetic samples $M = K \times J$
> >    Scores $\leftarrow 0^M$
> > 2. $\text{Scores}[j] = \frac{1}{|E_i|} \sum_{e_{pri} \in E_i} \frac{\langle e_{pri}, e_j \rangle}{\|e_{pri}\| \|e_j\|}$   for $e_j \in E_{syn}$
> > 3. **return** $Scores/\|Scores\|$ if $\|Scores\| > 1$ else Scores
>
> We switched from nearest-neighbor (NN) to cosine similarity primarily performed better. As shown in this experiment on the bioRxiv dataset with full participation (https://imgur.com/a/pg1H5iq), using cosine similarity results in a significantly lower FID score compared to the NN histogram. This improvement is because NN histograms produce sparser scores—often assigning zero to all synthetic samples associated with a given prompt in POPri. The chosen and rejected samples end up being essentially random. By contrast, cosine similarity provides more informative and continuous scoring.
>
>
> On privatization: we use the gaussian mechanism (line 17) where clients add noise. Their outputs are securely aggregated [3] before being given to the server. This gives the user-level distributed DP guarantee mentioned earlier.
>
>
> [3] Bonawitz et al, “Practical Secure Aggregation for Privacy-Preserving Machine Learning” https://eprint.iacr.org/2017/281.pdf
>
> **However, the approach appears to be a straightforward extension of private evolution algorithms**
>
> Thank you for your comment. You are correct that POPri is an extension of PE. However, we believe that properly re-interpreting PE as a preference optimization problem is not obvious a priori. In order to do so, we had to make careful design decisions such as (1) what client feedback we should collect and (2) how we should use the client feedback. This included carefully organizing the synthetic samples and their feedback to ensure that preference datasets could be built, and experimenting/reasoning over what LLM fine-tuning method to use (DPO, SFT, IPO).
>
> **I believe it is essential to include a discussion on the overall computational complexity of the proposed method (both client and server sides) relative to existing approaches.**
>
> This is a valid concern. The computational cost of POPri is indeed significantly larger than PE on the server side. Below, we provide a table on the server and client compute cost breakdowns when we sample 1000 clients per round on the bioRxiv dataset:
>
> | Method   | Client Runtime in GPU seconds | Server Runtime in GPU seconds     |
> |-|---|--|
> | PrE-Text | 0.0027 sec                 | 326.25 sec                      |
> | POPri    | 0.01 sec                   | 13,547.84 sec |
> | FedAvg   | 4.8 sec                    | -                               |
>
> The client compute cost of POPri and PrE-Text are both small compared to FedAvg. However, POPri has the highest server compute cost. POPri is suitable in cases where (1) we care a lot about the downstream model quality, (2) the client computation and/or communication is expensive, and (3) server computation is relatively cheap and powerful. We will add this explanation and data to the main paper.
>
> **In Line 20 of Algorithm 1, why does the server aggregate the scores from all clients instead of the clients in S^t**
>
> This is a typo, thank you for noticing. The server aggregates scores only from clients in S^t. We will fix it!

---

### Official Review · Reviewer_rhem · 2025-03-19

**Overall Recommendation:** 3

**Summary:**

In the paper, the authors present a novel approach to improving the utility of differentially private federated learning (DP-FL) by leveraging preference-optimized synthetic data generated through large language models (LLMs). The proposed method aggregates client feedback into preference pairs, and then fine-tunes an LLM to generate high-fidelity synthetic data while maintaining differential privacy (DP) guarantees.

**Claims And Evidence:**

The claims mentioned in the paper are all supported either theoretical or experimentally.

**Essential References Not Discussed:**

DP Model Parameters type methods:

[2] Yae Jee Cho, Luyang Liu, Zheng Xu, Aldi Fahrezi, and Gauri Joshi. 2024. Heterogeneous LoRA for Federated Fine-tuning of On-Device Foundation Models. In Proceedings of the 2024 Conference on Empirical Methods in Natural Language Processing, pages 12903–12913, Miami, Florida, USA. Association for Computational Linguistics.

[3] FDLoRA: Personalized Federated Learning of Large Language
Model via Dual LoRA Tuning, https://arxiv.org/pdf/2406.07925

**Experimental Designs Or Analyses:**

1. Similar to Point 2, can the proposed method handle the data with multiple clusters?

2. In figure 3, why the accuracy suddenly drops when number of clients per round reaches $10^4$ and the total number of clients is larger than $10^5$ in Congressional Speeches dataset?

3. Regarding baselines: the authors only includes the work DP Model Parameters type baselines, DP-FedAvg and DP-RFRL are works in 2016 and 2017, and the authors should include more recent works such as methods in [2], [3]

[2] Yae Jee Cho, Luyang Liu, Zheng Xu, Aldi Fahrezi, and Gauri Joshi. 2024. Heterogeneous LoRA for Federated Fine-tuning of On-Device Foundation Models. In Proceedings of the 2024 Conference on Empirical Methods in Natural Language Processing, pages 12903–12913, Miami, Florida, USA. Association for Computational Linguistics.

[3] FDLoRA: Personalized Federated Learning of Large Language
Model via Dual LoRA Tuning, https://arxiv.org/pdf/2406.07925

**Methods And Evaluation Criteria:**

Regarding the methods, my concerns are that:

1. Notations in Algorithm 2 are confusing. What’s $e_{pri}$? Also in the line 2 of the Algorithm 2, the expression of calculating histogram is hard to understand, and in the paper, there are no descriptions about the details of this line.

2. From figure 4, the authors only visualize when the evaluation dataset only has one cluster. How about multiple clusters? Can the proposed method can handle data with multiple clusters?

3. In section 5.2, the paragraph about the data distribution evolution, the authors mentions that early stopping can help avoids the overfitting. However, the details about early stopping are missing.

4. Regarding DPO:  the authors found that directly use rank 2 response as the rejected samples is not as well as the middle-ranked samples because of DP noise swapping ranking. In the preference learning literature, there are some existing works aiming to handle such pairwise noise, for example, the Distributionally robustifying DPO (Dr. DPO)[1]. I am just curious about whether such DPO method will work when taking rank 2 samples as the rejected samples.

[1] Towards Robust Alignment of Language Models: Distributionally Robustifying Direct Preference Optimization. https://arxiv.org/pdf/2407.07880

**Other Comments Or Suggestions:**

Typo: in the past paragraph of section 2.1, “Goal” -> “Goal.”

**Other Strengths And Weaknesses:**

Strengths

1. The authors provide a comprehensive related work.

2. The authors give a comprehensive ablation study.

**Questions For Authors:**

See point 1, 2, and 3 in Methods And Evaluation Criteria part

**Relation To Broader Scientific Literature:**

1. Replacing heuristic PE with preference optimization for higher-quality DP synthetic data.

2. Introducing LargeFedBench , a scalable, uncontaminated benchmark addressing gaps in prior datasets.

**Theoretical Claims:**

The authors provide the privacy guarantee of the proposed method.

---

> ### Author Rebuttal · Authors · 2025-04-01
>
> **Notations in Algorithm 2 are confusing. What’s e_pri? Also in the line 2 of the Algorithm 2, the expression of calculating histogram is hard to understand, and in the paper, there are no descriptions about the details of this line.**
>
> First, we would like to apologize for an oversight; Algorithm 2 is not the one actually used in our experiments. In POPri the clients actually score each synthetic sample based on the average cosine similarity between the synthetic embedding and their private data embeddings. The experimental results reported in the paper use this cosine similarity scoring method. Updated pseudocode for Algorithm 2 is below (which explains what e_pri is):
>
> >
> > 1. **Input:** Embeddings of private client data $E_i$ for $i \in S^t$, embeddings of synthetic data $E_{syn}$, total synthetic samples $M = K \times J$
> >    Scores $\leftarrow 0^M$
> > 2. $\text{Scores}[j] = \frac{1}{|E_i|} \sum_{e_{pri} \in E_i} \frac{\langle e_{pri}, e_j \rangle}{\|e_{pri}\| \|e_j\|}$   for $e_j \in E_{syn}$
> > 3. **return** $Scores/\|Scores\|$ if $\|Scores\| > 1$ else Scores
>
>
> We switched from nearest-neighbor (NN) to cosine similarity primarily performed better. As shown in this experiment on the bioRxiv dataset with full participation (https://imgur.com/a/pg1H5iq), using cosine similarity results in a significantly lower FID score compared to the NN histogram. This improvement is because NN histograms produce sparser scores—often assigning zero to all synthetic samples associated with a given prompt in POPri. The chosen and rejected samples end up being essentially random. By contrast, cosine similarity provides more informative and continuous scoring.
>
> [1] Lin et al, “Differentially private synthetic data via foundation model APIs 1: Images”, ICLR 2023
>
>
> **From figure 4, the authors only visualize when the evaluation dataset only has one cluster. How about multiple clusters? Can the proposed method can handle data with multiple clusters?**
>
> Thank you for the question. Figure 4 shows the bioRxiv dataset, which is only one of our benchmark datasets. Our Congressional Speeches dataset, which contains congressional speeches from the United States, Canada, and the United Kingdom, contains multiple clusters by country: (see t-SNE plot https://imgur.com/a/IVKgbDL). Given that POPri performs the best on Congressional Speeches as well, our experiments do suggest that POPri can effectively handle data with multiple clusters.
>
> **However, the details about early stopping are missing.**
> In Appendix B.1, we explain our early stopping criterion. Note that our FedAvg and DP-FTRL baselines also engage in the same form of early stopping (using validation set loss) for fair comparison, and PE (and PE+SFT) did not benefit from stopping early in our experiments.
>
>
> **I am just curious about whether such DPO method will work when taking rank 2 samples as the rejected samples.**
>
> Thank you for the great suggestion! Dr. DPO is a strong work tackling the pairwise noise issue in DPO, and we will discuss it in the paper. We are working on an experiment comparing against cDPO (a version of DPO robust to label noise, https://ericmitchell.ai/cdpo.pdf), and will update it here. We also already compare against IPO (Gheshlaghi Azar et al 2023, https://arxiv.org/abs/2310.12036) in Table 3 of the appendix, which also prevents overfitting to incorrect labels (see https://ericmitchell.ai/cdpo.pdf), and found that it performed worse than DPO.
>
> **In figure 3, why the accuracy suddenly drops when number of clients per round reaches 10^4**
>
> In federated learning, past work has shown that sampling more clients per round can reduce generalization performance (see Figure 3 in https://arxiv.org/abs/2106.07820, NeurIPS 2021). The authors of that work believe that the “number of clients per round” parameter is similar to batch size in centralized training, where larger batch size can lead to worse generalization (https://arxiv.org/abs/1609.04836, ICLR 2017).
>
> **Regarding baselines: the authors only includes the work DP Model Parameters type baselines, DP-FedAvg and DP-RFRL are works in 2016 and 2017, and the authors should include more recent works such as methods in [2], [3]**
>
> LoRA (used in [2]) methods generally are designed to improve the efficiency of learning, not the final model quality. LoRA (and other parameter-efficient fine-tuning variants) generally do not outperform full fine-tuning. In [2], their method performs worse than FedAvg (Full) in Table 3. [3] is out of scope for our paper because they study a setting where each client can have a separate model, whereas our setting aims to train a single model. Hence, we do not believe that either of these papers should be included as baselines. Moreover, we do not know of any algorithms that outperform DP-FTRL in the setting we are considering.
>
> We thank the reviewer for their time in helping improve our paper, and hope that our responses addressed the reviewer’s concerns!

---

> > ### Comment · Reviewer_rhem · 2025-04-07
> >
> > The authors' response has addressed my concern, and I raise my score to 3.

---

> > > ### Author Response · Authors · 2025-04-07
> > >
> > > Thank you!
> > >
> > > In our original rebuttal we mentioned that we would run a comparison against cDPO to see if it could help address the label noise issue. The experiment setting is as follows: we used the bioRxiv dataset, set eps=7, learning rate = 8e-7, and used the third-ranked sample (in our experiments using the second-ranked sample did not work regardless of noise setting, possibly due to the two samples being too difficult to distinguish from each other).
> > >
> > > We present our results here: https://imgur.com/a/xamA7yZ
> > >
> > > We find that using cDPO with a label_smoothing parameter to 0.02 and 0.05 improved the FID of the synthetic samples, with 0.02 performing better than 0.05. The reviewer's intuition was correct--using robust versions of DPO can improve the results by making us more resilient to label noise.

---

### Official Review · Reviewer_Yas7 · 2025-03-23

**Overall Recommendation:** 4

**Summary:**

This paper proposes a method for differentially private federated learning of language data, which finetunes a pretrained LLM with synthetic data generated according to client preferences. The method is lightweight in terms of client computation and client-server communication, achieves guarantees of differential privacy, and experimentally outperforms baselines. Ablation studies are also included.

## Update after rebuttal
Discussion with the authors helped clarify a few points (choice of DP parameter $\delta$, relative communication cost of different algorithms) and I will maintain my recommendation of acceptance.

**Claims And Evidence:**

All of the claims in the paper are backed up by theoretical and experimental evidence. I did not find any problematic claims.

**Essential References Not Discussed:**

I am not aware of any essential references not discussed.

**Experimental Designs Or Analyses:**

I am very satisfied with the experimental design of the paper. The experiments compare against a number of relevant baselines, report error bars, carefully handle data contamination, and run interesting ablation studies.

**Methods And Evaluation Criteria:**

The evaluation methods make sense for the problem at hand. The authors introduced a dataset of federated language data, with special attention paid to avoid any data contamination, which brings rigor to the experiments and is a contribution in its own right.

**Other Comments Or Suggestions:**

One small formatting suggestion. For the equation on line 240, column 2, I recommend to use \left( and \right) for parentheses, so that the parantheses are not smaller than the expressions inside of them.

**Other Strengths And Weaknesses:**

Strengths
1. The proposed algorithm outperforms baselines pretty convincingly. The gap is not huge (between 1 and 2 absolute percentage points improvement to next-token prediction accuracy), but the error bars are non-overlapping, so I think that the improvement is reliable.
2. The experimental design is rigorous. The authors report error bars, and they introduce their own dataset to eliminate the possibility of data contamination. I also appreciate the ablation studies, which demonstrate robustness to the number of participating clients and probe the design choices of the proposed algorithm. In particular, I appreciate the ablation study in Figure 4, which uncovers a potential weakness of the proposed algorithm.
3. The paper is very well written. All of the technical ideas are well communicated, all of the baselines and potential variations of the problem setup are explained well, and the novelty of the proposed work is clearly distinguished. The authors are also very transparent about the possible limitations of their work, which I appreciate.

Weaknesses
1. The technical novelty of the paper is somewhat incremental. To my understanding, the algorithm is essentially an extension of Private Evolution (PE), but clients provide feedback on synthetic data through preference ranking instead of by choosing the best samples. Still, the performance of the proposed algorithm is convincing enough that I think this is a minor weakness.

**Questions For Authors:**

1. Can you explain the choice of $\delta$ given in the paragraph starting on line 376, column 1? I'm curious whether the gap between the proposed algorithm and baselines might grow or shrink depending on the choice of $\delta$. Did you ever perform any evaluations with different choices of $\delta$?
2. Can you elaborate on the "Cost analysis" in Section 5.1? I'm confused about why POPri communicates 10x more embeddings than PE methods. Is this because of a fundamental difference in the algorithms, or is the number of communicated embeddings a parameter that you can freely choose? In the latter case, is it possible that the improved performance of PoPri compared to PE methods is caused by the larger number of communicated embeddings? Did you ever compare against PE methods while controlling for the communication cost? I want to clarify that this is not a major issue, as it seems the current experiments control for the privacy budget instead of the communication budget, which is reasonable. Still, I want to understand where this 10x difference comes from.

**Relation To Broader Scientific Literature:**

This paper contributes to the practice of federated learning with provable guarantees of privacy, particularly in the context of training language models. The proposed method appears very practical for use with edge devices (e.g. mobile phones), since the computation and communication costs for the clients is very small compared with classic FL methods. This line of work has important societal implications, since large tech companies can leverage existing infrastructure to immediately implement and deploy these kinds of algorithms with user devices and user data.

**Theoretical Claims:**

There are no proofs in the paper. The only theory-adjacent claims are the ones related to differential privacy guarantees, but these guarantees are essentially handled by the Opacus library.

---

> ### Author Rebuttal · Authors · 2025-04-01
>
> **The technical novelty of the paper is somewhat incremental. To my understanding, the algorithm is essentially an extension of Private Evolution (PE), but clients provide feedback on synthetic data through preference ranking instead of by choosing the best samples. Still, the performance of the proposed algorithm is convincing enough that I think this is a minor weakness.**
>
> Thank you for your comment. You are correct that POPri is an extension of PE. However, we believe that re-interpreting PE as a preference optimization problem is not obvious a priori.
>
> PE was initially presented as an alternative to fine-tuning synthetic data models. In [1], the authors argue that PE can be used to generate DP synthetic data when fine-tuning a model is computationally impractical, but one has access to a foundation model API. Their key insight was that in this setting, one can obtain high-quality DP synthetic data by converting the synthetic data generation problem from a DP optimization problem to an iterative private selection problem, which is much lower-dimensional than private optimization. In particular, the private selection problem aims to select a single item from a set with the highest reward (according to a dataset-specific score function), subject to a DP constraint. Since [1] did not need to fine-tune the base foundation model, they were able to get away with using extremely coarse-grained feedback from the clients—namely, private selection data.
>
> Our paper adds important nuance to this understanding of private synthetic data by arguing that even fine-tuning problems benefit from asking clients to solve a private selection problem. How? By not viewing clients’ private selections as “selections” at all, but as preferences. Indeed, this perspective motivates several design choices in POPri (for example, generating J samples from each of K random prompts to enable the construction of a preference dataset) which are essential for making our method outperform PE. As you observed, our empirical results improve significantly as a result of this interpretation.
>
> [1] Lin et al, “Differentially private synthetic data via foundation model APIs 1: Images”, ICLR 2023
>
> **Can you explain the choice of $\delta$ given in the paragraph starting on line 376, column 1? I'm curious whether the gap between the proposed algorithm and baselines might grow or shrink depending on the choice of $\delta$. Did you ever perform any evaluations with different choices of $\delta$?**
>
> This is a good question. While we did not directly conduct ablations with different values of delta (due to computational limits and time constraints for the rebuttal), we did run experiments for different values of epsilon. Because the (epsilon,delta) values are computed using a privacy accounting mechanism, we can equivalently interpret the results from (epsilon=7, delta=1e-6) (Appendix E, Figure 6) as a smaller epsilon with a larger delta. For example, under the same noise and number of rounds, the result from (Appendix E, Figure 6) is also (epsilon=1, delta=0.45). Under this interpretation, we can directly compare Figures 3 and 6, as keeping epsilon constant and varying delta. Since the results change minimally across plots, our intuition is that the results are not very sensitive to delta, even across large differences. We will add a full ablation in the final version where we vary delta explicitly to verify this intuition.
>
> **Can you elaborate on the "Cost analysis" in Section 5.1? I'm confused about why POPri communicates 10x more embeddings than PE methods. Is this because of a fundamental difference in the algorithms, or is the number of communicated embeddings a parameter that you can freely choose? In the latter case, is it possible that the improved performance of PoPri compared to PE methods is caused by the larger number of communicated embeddings? Did you ever compare against PE methods while controlling for the communication cost? I want to clarify that this is not a major issue, as it seems the current experiments control for the privacy budget instead of the communication budget, which is reasonable. Still, I want to understand where this 10x difference comes from.**
>
> Thank you for this great suggestion. We have not run experiments in which we control the communication cost. The number of communicated embeddings is a number we can freely choose (K x J), and we can choose K and J. The reason the communication is 10x is because we set K to be the same as N_syn in PE so that the POPri preference dataset had the same size as the PE iterative synthetic dataset, and set J to be 10 because that number worked well among different choices we made (in early experiments we had tried 2, 4, 6, 8, 10).
>
> We thank the reviewer for their time in helping improve our paper!

---

> > ### Comment · Reviewer_Yas7 · 2025-04-03
> >
> > Thank you for your response. I think that including a full ablation around epsilon and delta will definitely improve the paper. My concern still holds regarding the limited technical novelty, but I think that the results are strong enough to justify acceptance. I will keep my score the same.

---

### Decision · Program_Chairs · 2025-05-01

**Decision:**

Accept (poster)

**Comment:**

This paper proposes a method called POPri, for federated learning with differential privacy guarantees in the context of language models. In addition, this paper also creates a new benchmark for evaluating federated learning in the era of large language models. Most reviewers are positive about this paper, but reviewer W8ui was concerned about the limited downstream task evaluation and the low accuracy. Therefore I recommend a weak acceptance of this paper.